# LEARNED DATA TRANSFORMATION: A DATA-CENTRIC PLUGIN FOR ENHANCING TIME SERIES FORECASTING

## ABSTRACT

Data-centric approaches in Time Series Forecasting (TSF) often involve heuristic-based operations on data. This paper proposes to find a general end-to-end data transformation that serves as a plugin to enhance any arbitrary TSF model's performance. Our idea is to generate transformed data during an approximating process and to co-train a predictor for evaluating data with the transformation. To achieve this, we propose the Proximal Transformation Network (PTN), which learns effective transformations while maintaining proximity to the raw data to ensure fidelity. When orthogonally integrated with popular TSF models, our method helps achieve state-of-the-art performance on seven real-world datasets. Additionally, we show that the proximal transformation process can be interpreted in terms of predictability and distribution alignment among channels, highlighting the potential of data-centric methods for future research. Our code is available at https://anonymous.4open.science/r/PTN-2FC6/.

## 1 INTRODUCTION

Time Series Forecasting (TSF) is a significant task in various domains including energy, finance, engineering, and industrial applications. Recently, along with the tremendous achievements of neural networks in deep learning, TSF has seen many learning-based methods that consistently outperform traditional methods in terms of precision, robustness, and capacity (Zhou et al., 2021; Wu et al., 2021; Salinas et al., 2020; Taylor & Letham, 2018).

Meanwhile, modeling specific time series data properties such as distribution shift (Kim et al., 2021), non-stationarity (Liu et al., 2022), frequency bias (Piao et al., 2024), and variate covariance (Wang et al., 2024c) has achieved outstanding successes in TSF. The successes can be attributed to sharp observations of commonalities in time series and sound designs from a model-centric perspective. With the angle shifted, data-centric methods for time series including normalizations (Fan et al., 2023; Han et al., 2024; Deng et al., 2021; Liu et al., 2024c; September et al., 2024), augmentations (Zheng et al., 2024), and simple preprocessing methods (e.g., down-sampling (Yu et al., 2023; Wang et al., 2024a) and patching (Nie et al., 2023; Wang et al., 2024b)) can improve the overall performance of a given model by adopting data-level operations. These methods either explicitly or implicitly entail a specific transformation applied to the raw time series data. This paper further pushes the boundary of data-centric methods by answering the following: *Is there a general data transformation that can improve a model's generalization capability? If so, can we discover it in an end-to-end manner?*

We first verify the insufficiency of specific designs to generalize on data with different complexities. The evidence is provided by an example that involves learning on noisy input-target pairs generated by a linear mapping. When smoothing the noised data is the only transformation to consider, linear mapping with different complexities has different best-denoised data selected from a range of common denoising/smoothing methods. We then reformulate our problem to identify a general process, which finds an effective transformed data given any TSF model to achieve better forecasting results than if using the raw time series data directly. We refer to this problem as finding the optimal transformed data for forecasting, which extends the original optimization space by adding another dimension w.r.t. data fidelity. However, finding such a transformation is challeng-

ing. Assuming the proximal existence of our objective, we constrain the problem as a simplified co-optimization on the proximity of the transformed data and forecasting accuracy for each level of proximity. The transformation is provided by a Convolution Encoder, and an approaching process is learned by an attention-based Decoder. Combining these two components, we propose the **P**roximal **T**ransformation **N**etwork (PTN) that approximates the raw data at different distances to find the transformed data that results in better performance for a given model.

In summary, our contributions lie in the following aspects:

- *A Reformulated Problem.* We consider the conventional TSF problem as a two-step problem: finding a data transformation that enables better generalization and performance, followed by learning to forecast on the transformed data.
- *A Plug-and-Play Module.* We propose the Proximal Transformation Network to find an effective data transformation. The model includes a convolution-based Encoder and an attention-based Decoder that provide transformation on different levels of proximity. The training involves a co-optimization of the proximity of the transformed data and forecasting accuracy.
- *Extensive Experiments on Real-World Datasets.* By orthogonally adding our module to several representative models, we managed to achieve state-of-the-art (SOTA) performance on seven real-world datasets. The Proximal Transformation can be further interpreted by predictability assessments and distribution alignments.

## 2 RELATED WORK

We focus on data-centric aspects within TSF that transform data explicitly or implicitly. A discussion of TSF models based on their designs and advances is presented in Appendix A.2.

**Normalization**   Normalization is crucial for neural networks on time series data (Bhanja & Das, 2019). Techniques like Reversible Instance Normalization (Kim et al., 2021) address instance-level distribution shifts, influencing models like the non-stationary Transformer (Liu et al., 2022). Extensions include local and global normalizations (Fan et al., 2023; Han et al., 2024), and channel-wise normalization that aligns multivariate series via adaptive modules (Deng et al., 2021; Zhao & Shen, 2024). Such adaptive normalizations enhance expressiveness with additional parameters (Liu et al., 2024c; September et al., 2024).

**Data Augmentation**   Traditional data augmentations (e.g., cropping, jittering, and scaling (Wen et al., 2020)) improve time series representation learning (Yue et al., 2022; Fan et al., 2020). InfoTS (Luo et al., 2023) is a novel proposed meta-learning-based method that learns an end-to-end augmentation from a predefined set of transformations for Contrastive Learning. AutoTCL (Zheng et al., 2024) further offers theoretical insights based on Information Bottleneck Theory (Tishby et al., 2000) and demonstrates effectiveness on CoST (Woo et al., 2022). These adaptive data augmentations still explicitly rely on specific operations, unable to provide a general data transformation.

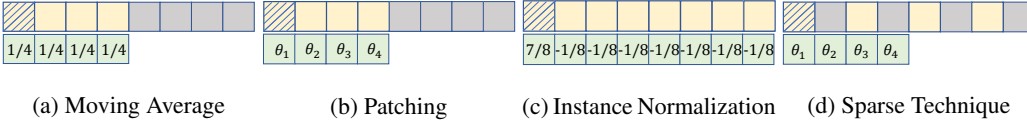

(a) Moving Average    (b) Patching    (c) Instance Normalization    (d) Sparse Technique

Figure 1: Four typical types of **Moving Kernel Smoothing** methods in TSF models (yellow grids indicate the elements involved in kernel computation, shadowed grids indicate the position to place the result): **(a)** *Moving Average* that computes average value within a kernel window (Zeng et al., 2023); **(b)** *Patching* that computes a learnable weighted sum within the kernel (Nie et al., 2023); **(c)** *Instance Normalization* that substracts the mean value of the given series (Kim et al., 2021; Li et al., 2023), $\frac{n-1}{n}$ for the centered weight and $-\frac{1}{n}$ for the rest; and **(d)** *Sparse Techinque* that computes weighted average at a defined stride (Lin et al., 2024).

**Moving Kernel Smoothing**   Several methods function as smoothing techniques that inherently transform data (see Figure 1). DLINEAR (Zeng et al., 2023) uses moving average decomposition with linear models to outperform Transformers. RLINEAR (Li et al., 2023) simplifies this with instance normalization, achieving nearly SOTA performance. The patching operation (Nie et al.,

2023) effectively aggregates features, inspiring works on patch-level features (Wang et al., 2024b; Tang & Zhang, 2024). Recently, SPARSETSF (Lin et al., 2024) introduces a sparse downsampling technique (Yu et al., 2023; Wang et al., 2024a), matching SOTA models with only 1k parameters.

Most of these works have not been applied to recent models, and designs like channel-wise and multi-level normalizations are unorthogonal to some backbones. The most comparable work is AutoTCL (Zheng et al., 2024). However, unlike AutoTCL, our method does not heavily emphasize strong theoretical guarantees and focuses on supervised learning (instead of contrastive learning) that typically delivers superior performance in TSF. Moreover, our method is designed as an end-to-end data transformation framework, encompassing all three key aspects mentioned above.

## 3 ANALYSIS ON LEARNED DATA TRANSFORMATION

### 3.1 MOTIVATION EXAMPLE

**Data Synthesis** *Learning a linear mapping* with a *linear model* from $(X, Y)$ data pairs generated by the same linear mapping would be a good example to gain insights into effective data transformations. To make a meaningful example, we introduce independent and identically distributed (IID) **noise** and apply typical denoising methods, as depicted in Figure 1. Generally, these practices offset IID noise by computing weighted averages of data points within moving kernels. Back to our example, we generate a synthetic toy dataset as follows.

$$Y = W(X + \epsilon_X) + \epsilon_Y \tag{1}$$

$$\text{s.t.} \quad |W|_1 \le k \tag{2}$$

where $X \in \mathbb{R}^{H \times N}$, $Y \in \mathbb{R}^{L \times N}$ where $L$ and $H$ are the series lengths and $N$ is the data point number; noises $\epsilon_X \in \mathbb{R}^{L \times N}, \epsilon_Y \in \mathbb{R}^{H \times N}$ are sampled from the Normal distribution $\mathcal{N}(\mu, \sigma)$; and $W$ is the weight matrix subject to an $\ell_1$-norm constraint.

In our synthesis, we represent the **complexity**[1] of the $X$-to-$Y$ mapping using the $\ell_1$-norm of $W$[2] and constrain it with a parameter $k$ (see Equation 2). To ensure that $X$ and $Y$ have equal means — thereby eliminating concerns about distribution shift — we apply Softmax to the last dimension of $W$. This allows us to focus on the impact of smoothing/denoising methods on the performance of linear models. We also introduce an ORACLE linear mapping method, using a hypothetically known $W$ to directly obtain $Y$.

Table 1: Typical linear models on the toy examples: (1) MALINEAR, a linear model applying Moving Average (Figure 1(a)) on $X$ only; (2) BIMALINEAR, a linear model applying Moving Average on both $X$ and $Y$; (3) PATCHLINEAR, using Patching (Figure 1(b)), PATCHTST with attention removed; (4) RLINEAR, using Instance Norm (Figure 1(c)), the exact copy of (Li et al., 2023); (5) SPARSETSF, using Sparse Technique (Figure 1(d)), the exact copy of (Lin et al., 2024); (6) ORACLE forecasting by $Y = WX$, presented as a relatively strong baseline to compare with.

| Models Smoothing | MALINEAR♣ Moving Average | | BIMALINEAR♣ Moving Average | | RLINEAR Instance Norm | | PATCHLINEAR Patching | | SPARSETSF Sparse Technique | | ORACLE | |
|---|---|---|---|---|---|---|---|---|---|---|---|---|
| $|W|_1$ | MSE | MAE | MSE | MAE | MSE | MAE | MSE | MAE | MSE | MAE | MSE | MAE |
| 0.0030 | 0.0290 | 0.1307 | 0.0275 | 0.1258 | 0.0119 | 0.0866 | 0.0103 | 0.0809 | **0.0101** | **0.0801** | 0.0101 | 0.0801 |
| 0.0052 | 0.0150 | 0.0970 | 0.0141 | 0.0943 | 0.0111 | 0.084 | 0.0108 | 0.0830 | **0.0101** | **0.0803** | 0.0101 | 0.0803 |
| 0.0104 | 0.0111 | 0.0841 | 0.0111 | 0.084 | 0.0106 | 0.0822 | 0.0108 | 0.0829 | **0.0103** | **0.0808** | 0.0103 | 0.0808 |
| 0.0143 | 0.0168 | 0.1035 | 0.0112 | 0.0844 | 0.0107 | 0.0826 | **0.0105** | **0.0818** | 0.0106 | 0.0821 | 0.0107 | 0.0826 |
| 0.0214 | 0.0169 | 0.1037 | 0.0114 | 0.0850 | 0.0108 | 0.0830 | **0.0106** | **0.0822** | 0.0112 | 0.0846 | 0.0117 | 0.0861 |
| 0.0437 | 0.0175 | 0.1056 | 0.0120 | 0.0872 | **0.0113** | **0.0847** | **0.0113** | 0.0850 | 0.0139 | 0.0941 | 0.0157 | 0.0997 |
| 0.0761 | 0.0148 | 0.0950 | 0.0142 | 0.0946 | **0.0127** | **0.0897** | 0.0132 | 0.0917 | 0.0229 | 0.1195 | 0.0265 | 0.1291 |

♣ Decomposition, from another angle, interprets Moving Average results as Trend term and the residual as Seasonal term. In our toy example with synthesized noise, the DLINEAR model, which also models the Seasonal term, tends to overfit the noise. We thus use MALINEAR and BIMALINEAR, modified versions of DLINEAR that focus solely on the Trend term for denoising. In the main experiment (Section 5), we use DLINEAR instead of the other two for brevity, since both terms are meaningful in real scenarios.

**Observations** According to Table 1, our experiments with toy datasets using various linear models and smoothing methods reveal that different smoothing methods achieve the best performance de-

---

[1]A relevant notion is sparsity (Lin et al., 2024); we provide more discussions in Appendix A.7.1.

[2]We use the convex and computationally more tractable $\ell_1$-norm instead of the $\ell_0$-norm (Tibshirani, 1996).

pending on the mapping complexities of the datasets. Since all models use the same one-layer linear backbone, performance differences are solely due to smoothing methods.

The performance variation related to the $X$-to-$Y$ mapping complexity indicates that choosing the right data transformation is crucial. However, this complexity information becomes available only after an approximate training on $(X, Y)$ is completed. For problems with intricacy beyond this toy example, it is less likely to acquire the priors on the underlying data patterns. Additionally, for Patching and Sparse Technique (5th and 6th columns), hyperparameters regarding their kernel designs — patch length (Nie et al., 2023) and period length (Lin et al., 2024), respectively — are often sensitive to different datasets. Therefore, it is essential to **develop transformations that are learnable and general**, enabling them to adapt to any data and model without relying on heuristics. Another observation is that BIMALINEAR consistently outperforms MALINEAR (2nd and 3rd columns), suggesting that **using a uniform transformation for both $X$ and $Y$ is more effective**.

## 3.2 TSF PROBLEM REFORMULATION

In many previous studies, TSF follows a paradigm that aims to find the direct mapping from the most recent look-back window to the target series. Let the context series (*resp.* target series) be $X \in \mathbb{R}^{L \times N}$ (*resp.* $Y \in \mathbb{R}^{H \times N}$) with time series length $L$ (*resp.* $H$) and the data point number $N$. Without loss of generality, we only consider the univariate scenario in our intuitive reformulation.

**Original Problem.** *Find the optimal $\theta$ minimizing training error:* $\theta^* = \arg\min_{\theta} \|f(X; \theta) - Y\|$, *using a specified error metric $\|\cdot\|$, typically the $\ell_1$- or $\ell_2$-norm.*

As noted in Section 3.1, transformations like smoothing vary in optimality across datasets. We thus propose a more general, learnable data transformation that exists for any data and predictor model.

**Proposition 1.** *Given any time series segmented into $(X, Y)$, and a predictor model $f(\cdot; \theta)$ where $\|f(X_{test}; \theta^*) - Y_{test}\| > 0$, with $\theta^* = \arg\min_{\theta} \|f(X_{train}; \theta) - Y_{train}\|$, there exists a transformation $(\tilde{X}, \tilde{Y})$ such that $\left\|f(X_{test}; \tilde{\theta}^*) - Y_{test}\right\| < \|f(X_{test}; \theta^*) - Y_{test}\|$, where $\tilde{\theta}^* = \arg\min_{\theta} \left\|f(\tilde{X}_{train}; \theta) - \tilde{Y}_{train}\right\|$.*

In real scenarios, this proposition holds as long as there exists a distribution difference/shift in raw series, which is common in well-recognized TSF datasets (Kim et al., 2021). We showcase that the better $\tilde{X}$ and $\tilde{Y}$ exist in a simple scenario for linear models in Appendix A.1. When Proposition 1 holds, we consider the following reformulation of the problem:

**Reformulated Problem.** *Find a transformation $\mathcal{G}(X; \phi^*)$, such that*

$$\phi^* = \arg\min_{\phi} \|f(X; \theta^*) - Y\|, \text{ where } \theta^* = \arg\min_{\theta} \|f(\mathcal{G}(X; \phi); \theta) - \mathcal{G}(Y; \phi)\|.$$

## 3.3 LOSS SURFACE TRANSFER IN THE REFORMULATED PROBLEM

Our study seeks a validated transformation $\tilde{X} = \mathcal{G}(X; \phi)$ that enhances the predictor $f(\cdot; \theta)$'s generalization and robustness to varying mapping complexity in $(X, Y)$. Searching for arbitrary transformations can be tedious, and we have no clue about what loss to use to guide such transformation. To address this, we constrain the transformation to be "somewhere near" the raw data, leading to a co-optimization process: learn prediction on the transformed data and align it with the raw series.

We use stochastic gradient descent on a combination of proximity and prediction losses. The **proximity loss** measures the difference between the transformed and raw data, while the **prediction loss** assesses prediction accuracy:

$$\mathcal{L}_{pareto} = \mathcal{L}_{prox} + \mathcal{L}_{pred} = \left(\left\|\tilde{X} - X\right\| + \left\|\tilde{Y} - Y\right\|\right) + \left\|f(\tilde{X}; \theta) - \tilde{Y}\right\| \quad \text{s.t.} \quad \nabla f \leq \alpha, \quad (3)$$

where $\tilde{X} = \mathcal{G}(X; \phi)$ is transformed from the raw data via a network parameterized by $\phi$, and $\nabla f \leq \alpha$ is a convergence constraint on the predictor. When applying to a linear model, since the optimization of that model is convex, our constraint guarantees reaching the Pareto frontier.

To provide an intuitive demonstration, we employ the Proximal Transformation Network (PTN) (detailed soon) for the transformation and DLINEAR (Zeng et al., 2023) as the predictor that is

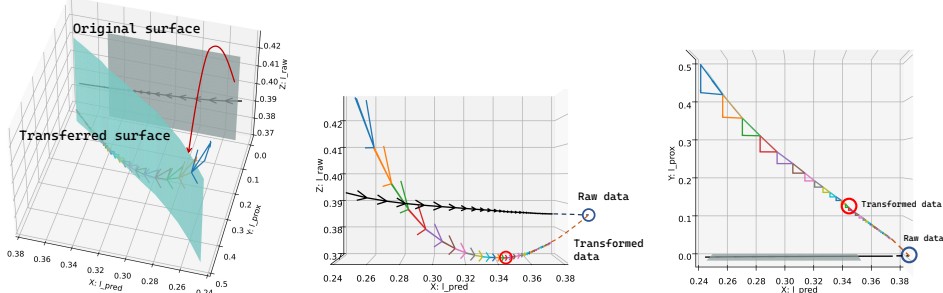

(a) Overview of surface transfer (b) Improved prediction on raw data (c) Pareto frontier w.r.t. surface

Figure 2: PTN with DLINEAR (colored curves) vs vanilla DLINEAR (black curves). The z-axis ($l\_raw$) refers to the forecasting accuracy measured by MSE on the raw test dataset; the x-axis ($l\_prox$) and y-axis ($l\_pred$) are $\mathcal{L}_{prox}$ and $\mathcal{L}_{pred}$, respectively, for the loss curve on our PTN. For the vanilla DLINEAR, $l\_pred$ refers to the MSE on the training set, and $l\_prox$ equals zero as it is trained on the raw data. **(a)** Loss surface transfer achieved by our Proximal Transformation; **(b)** MSE loss curve measured on the raw test set indicating that the best parameters achieved by PTN outperform the same parameters trained on the raw data; **(c)** Pareto frontier formed during training, illustrating the trade-off between proximity and prediction losses.

trained on the transformed data. The training process is showcased on the ETTh1 dataset with look-back length and prediction horizon both set to 96. Figure 2(a) illustrates that the training on the raw data occurs on the plane where the proximity loss $l_{prox} = 0$ (gray plane), while training on transformed data happens on a curved surface (cyan surface). Intuitively, what our method does is to transfer the original plane to this curved surface (indicated by the red arrow).

In the $xOz$ projection (Figure 2(b)), the minimum loss on the curved surface is clearly lower than on the original plane. This surface, achieved by the co-optimization of proximity and prediction losses, balances both objectives by ideally attaining the optimal prediction loss at each point. Hence, it can be regarded as representing the Pareto frontier, as shown in the $xOy$ projection (Figure 2(c)). Assuming the intersection of the two loss curves (shown as the dotted lines in Figures 2(b) and (c)), we find the endpoint of this frontier an equivalent point to the raw-data-guided training. By expanding the optimization from the $l_{prox} = 0$ plane to the Pareto frontier's curved surface, we discover a superior transformation. Once the global minimum is found on the validation set, we fix the optimized parameters and apply them to the test set, following standard machine learning practice. By analyzing the curves and surfaces, we demonstrate that our method not only maintains fidelity to the raw data but also leverages the transformation to achieve enhanced performance.

## 4 PROXIMAL TRANSFORMATION NETWORK

Given the preliminaries in Section 3.2, we decompose our design into two parts: the Proximal Transformation Network (PTN) that gradually approaches the raw series via a learnable Encoder-Decoder transformation, and an arbitrary predictor model that is orthogonal to PTN. We further introduce a Mixture-of-Experts design within PTN, enabling the cooperative transformation of data patches and providing a new approach to scaling. The full architecture is shown in Figure 3.

### 4.1 ENCODER-DECODER ARCHITECTURE

**Encoder** Typical smoothing methods mentioned in Section 3.1 all compute the weighted sum within a given moving kernel, similar to a convolution layer. We then consider an Encoder (Figure 3(b)) consisting of a convolutional network that is deeper than the smoothing methods with at most two layers. Inspired by MODERNTCN (Luo & Wang, 2024), we consider the Effective Receptive Field (ERF) an important factor in designing our module. Instead of using an extra-large kernel, we enlarge the number of kernels in the convolution layer and unfold the ($K^i \times L^i$)-sized output to a sequence of embeddings of length $L = L^i \times K^i$, where $K^i$ is the number of kernels of the $i$-th

Figure 3: Architecture of PTN. The raw data $X$ and $Y$ are first fed into our proposed module PTN to generate transformed data $\tilde{X}$ and $\tilde{Y}$ (shown in blue arrows), then the Predictor is trained on the transformed data (shown in the green arrow). **(a)** Overview design with MoE; **(b)** Convolution Encoder generates embedding by concatenating outputs from various depths; **(c)** Intra-patch Attention computes attention on embedding with subseries; **(d)** Channel-wise Attention computes attention on specific channels; **(e)** Point-wise Linear Head shares parameters for all embeddings.

convolution layer, $L^i$ is the length of output from the $i$-th layer, and $L$ is the original input length. This includes a transpose-unfold operation, shown in detail in Figure 12 in Appendix A.4.5.

By setting the 1D convolution with kernel size = 3, stride = 1, and padding = 1, we can halve the input length at each layer. Doubling the number of kernels for each subsequent layer ensures that the receptive field width at the $i$-th layer is $(2^{i+1} - 1)$. This design can help generate a larger ERF as discussed in Appendix A.4.5. We believe that our design is more expressive and can serve as a uniform transformation initializer. The case study in Section 5.2 demonstrates that this module adaptively aligns the data distributions of variates in time series.

**Decoder** We introduce two attention mechanisms and a linear projection head (Figures 3(c)-(e)). ***Intra-patch attention*** computes attention scores within patches, while ***channel-wise attention*** computes attention scores across channels (here "channels" refer to variates instead of image channels).

$$\text{Intra-patch Attention: } \mathbf{e}_{\text{IP}} = \frac{\text{Softmax}\left((\mathbf{e}Q)\,(\mathbf{e}K)^{\top}\right)(\mathbf{e}V)}{\sqrt{d}}, \tag{4}$$

$$\text{Channel-wise Attention: } \mathbf{e}_{\text{Ch}} = \frac{\text{Softmax}\left((\mathbf{e}Q)\,(\mathbf{e}K)^{\top}\right)(\mathbf{e}V)}{\sqrt{d}}, \tag{5}$$

where $\mathbf{e}$ is the output embedding from Convolution Encoder, $\mathbf{e}_{\text{IP}} \in \mathbb{R}^{C \times L \times d}$ and $\mathbf{e}_{\text{Ch}} \in \mathbb{R}^{L \times C \times d}$ are the outputs of intra-patch attention and channel-wise attention, respectively, and $C$ denotes the number of variates. We add the two embeddings together as input of the next layer. The parameters of both attention mechanisms are shared for each layer to improve efficiency.

A ***linear head*** with a weight matrix $W_\theta$ of size $d \times 1$ (see Figure 3(e)) is shared for all data points for decoding, which is necessary for *point-wise decoding* from the latent embedding. Note that our approach differs from the common approach of concatenating all latent embeddings into a single long vector of length $L \times d$, which would require a large head of size $Ld \times L$. This is reasonable as the whole PTN module is designed to transform data and is not required to capture temporal dependencies for forecasting. *Point-wise decoding* can be also reinforced by our special design of Convolution Encoder, which ensures each embedding piece has a wide enough Effective Receptive Field (discussions extended in Appendices A.4.5 and A.4.6). Even with limited parameters, this module can explicitly decode the embedding, serving as the initial transformation for interpretability purposes. E.g., we can apply the same linear head to the latent embedding from Convolution Encoder to visualize what series is taken as the initialized transformation in PTN, as shown in Section 5.2.

**Training Loss** Using the Encoder-Decoder architecture, we obtain the transformed data $\tilde{X}$ and $\tilde{Y}$ in Equation 3. When paired with an arbitrary predictor, the model can be trained using this equation.

For more complex architectures, the gradient constraint in Equation 3 may hinder convergence, so we adopt a simplified Pareto Loss without it. Experimental results in Table 3 demonstrate that PTN is also effective when applied to Transformer architectures (Nie et al., 2023; Liu et al., 2024b).

## 4.2 MIXTURE OF EXPERTS FOR PATCHED SUBSERIES

One major concern with Time Series Transformers and other deep-learning-based TSF methods is the scaling law. Nie et al. (2023); Liu et al. (2024b) have shown that increasing the number of model parameters does not consistently improve performance. More often than not, these Transformers achieve their best results with as few as two layers.

We thus shift our approach to scaling by incorporating Mixture of Experts (MoE) into TSF. Specifically, we combine MoE with the patching operation, a common practice adopted by (Wang et al., 2024b; Zhou et al., 2023; Yu et al., 2023; Lin et al., 2023). In our design, we route patches to different expert models, each being a separate PTN with independent parameters, to enhance both performance and efficiency. Patches are routed to each expert based on their sequential order, as depicted in Figure 3(a). Detailed performance boost and a decomposition view of MoE are provided in Appendix A.5.

## 5 EXPERIMENTS

We evaluate the performance of our proposed PTN using seven widely recognized datasets: Electricity, Weather, Traffic, and four ETT datasets (ETTh1, ETTh2, ETTm1, ETTm2). These datasets are well-established benchmarks in the field and are publicly accessible (Wu et al., 2021).

We utilize iTRANSFORMER (Liu et al., 2024b), PATCHTST (Nie et al., 2023), TIMESNET (Wu et al., 2023), SPARSETSF (Lin et al., 2024), DLINEAR (Zeng et al., 2023), and FEDFORMER (Zhou et al., 2022) as our baselines, encompassing temporal Transformers, channel-wise Transformers, convolutional models, and linear models. Following previous settings (Zhou et al., 2021; Wu et al., 2021; 2023) for direct comparison, we set the forecasting horizon $H \in \{96, 192, 336, 720\}$ and look-back length $L \in \{96, 512\}$ for all datasets. We move the dataset descriptions, implementation details, and instructions for reproduction to Appendix A.3 and make the codebase accessible at https://anonymous.4open.science/r/PTN-2FC6/.

### 5.1 MAIN RESULTS

We address two main questions: **Q1**: Can PTN achieve SOTA results when paired with mainstream backbones without complex designs? **Q2**: Can PTN be applied to both linear models and non-linear Transformers to consistently improve performance?

For Q1, we use two representative forecasting models as backbones for PTN: RLINEAR (Li et al., 2023), a linear model only utilizing Instance Norm (Kim et al., 2021), and iTRANSFORMER (Liu et al., 2024b), a full Transformer architecture that inverts the feature dimension to compute attention scores across channels. We denote the combinations as PTN-RLI and PTN-ITR, respectively. Table 2 summarizes the average results for a look-back length of 96. Our models outperform others in 12 out of 14 evaluation metrics across seven datasets. Full results for look-back lengths of 96 and 512 are in Tables 11 and 12 in Appendix A.6. PTN-RLI leads on the four ETT datasets, where linear models typically excel, while PTN-ITR performs best on the other three datasets, aligning with iTRANSFORMER's SOTA results on large, multivariate datasets.

For Q2, we illustrate the performance boost from applying PTN to three linear models, including a basic single-layer linear model denoted as LINEAR. We also include two Transformer-based models: iTRANSFORMER (channel-wise) and PATCHTST (channel-independent). The results are presented in Table 3. Our PTN plugin consistently improves all five backbone models. For a simple LINEAR model enhanced with PTN, performance can even match SOTA baselines on some datasets. We omit the results for the other three ETT datasets and place their results in Table 13 in Appendix A.6.

Table 2: Average forecasting results of PTN-RLI and (PTN with RLINEAR and ITRANSFORMER backbones). The best and the second-best measures are in **bold** and underlined, respectively.

| Models | PTN-RLI | | PTN-ITR | | iTRANSFORMER | | PATCHTST | | TIMESNET | | SPARSETSF | | DLINEAR | | FEDFORMER | |
|---|---|---|---|---|---|---|---|---|---|---|---|---|---|---|---|---|
| Metric | MSE | MAE | MSE | MAE | MSE | MAE | MSE | MAE | MSE | MAE | MSE | MAE | MSE | MAE | MSE | MAE |
| ETTh1 | **0.437** | **0.427** | 0.453 | 0.451 | 0.454 | 0.448 | 0.469 | 0.455 | 0.458 | 0.450 | 0.445 | 0.425 | 0.446 | 0.434 | 0.440 | 0.460 |
| ETTh2 | **0.366** | **0.391** | 0.441 | 0.442 | 0.383 | 0.407 | 0.387 | 0.407 | 0.414 | 0.427 | 0.386 | 0.402 | 0.374 | 0.399 | 0.437 | 0.449 |
| ETTm1 | 0.399 | **0.390** | 0.394 | 0.396 | 0.407 | 0.410 | **0.387** | 0.400 | 0.400 | 0.406 | 0.414 | 0.392 | 0.414 | 0.408 | 0.448 | 0.452 |
| ETTm2 | **0.272** | **0.319** | 0.293 | 0.343 | 0.288 | 0.332 | 0.281 | 0.326 | 0.291 | 0.333 | 0.287 | 0.326 | 0.286 | 0.327 | 0.305 | 0.349 |
| Electricty | 0.215 | 0.289 | **0.164** | **0.253** | 0.178 | 0.270 | 0.205 | 0.290 | 0.193 | 0.295 | 0.221 | 0.291 | 0.219 | 0.298 | 0.214 | 0.327 |
| Traffic | 0.643 | 0.355 | 0.463 | **0.266** | **0.428** | 0.282 | 0.481 | 0.300 | 0.620 | 0.336 | 0.676 | 0.361 | 0.627 | 0.378 | 0.610 | 0.376 |
| Weather | 0.268 | 0.283 | **0.235** | **0.271** | 0.258 | 0.278 | 0.259 | 0.273 | 0.259 | 0.287 | 0.297 | 0.305 | 0.272 | 0.291 | 0.309 | 0.360 |

Table 3: Performance boost by adding PTN to different backbones with better results in **bold**.

| Models | LINEAR | | +PTN | | DLINEAR | | +PTN | | RLINEAR | | +PTN | | ITRANSFORMER | | +PTN | | PATCHTST | | +PTN | |
|---|---|---|---|---|---|---|---|---|---|---|---|---|---|---|---|---|---|---|---|---|
| Metric | MSE | MAE | MSE | MAE | MSE | MAE | MSE | MAE | MSE | MAE | MSE | MAE | MSE | MAE | MSE | MAE | MSE | MAE | MSE | MAE |
| ETTm1 | 0.400 | 0.398 | **0.394** | **0.389** | 0.414 | 0.408 | **0.401** | **0.390** | 0.405 | **0.390** | 0.399 | 0.390 | 0.407 | 0.410 | **0.394** | **0.396** | 0.387 | 0.400 | **0.381** | **0.386** |
| Electricity | 0.215 | 0.296 | **0.212** | **0.289** | 0.219 | 0.298 | **0.215** | **0.289** | 0.217 | 0.289 | **0.215** | **0.289** | 0.178 | 0.270 | **0.164** | **0.253** | 0.205 | 0.290 | **0.196** | **0.274** |
| Traffic | **0.626** | 0.363 | 0.675 | **0.352** | 0.627 | 0.378 | 0.643 | **0.355** | 0.672 | **0.354** | 0.643 | 0.355 | **0.428** | 0.282 | 0.463 | **0.266** | **0.481** | 0.300 | 0.584 | **0.294** |
| Weather | 0.265 | 0.295 | **0.263** | **0.292** | 0.272 | 0.291 | **0.268** | **0.283** | 0.284 | 0.287 | **0.268** | **0.283** | 0.258 | 0.278 | **0.235** | **0.271** | 0.259 | 0.273 | **0.246** | **0.272** |

## 5.2 INTERPRETATION OF PROXIMAL TRANSFORMATION

We shed some light on how and why the Proximal Transformation works. First, we show a self-supervised clustering pattern in the extended loss space for the transformed data. This pattern can be correlated to the predictability of time series, supporting the significant performance boosts from PTN. Second, we examine the training process of PTN, where the transformed data gradually approaches the raw data. In this process, the transformation adapts to channel-specific features related to data sparsity. Furthermore, we visualize the embeddings from our Convolution Encoder and observe that this design aligns distributions across channels, effectively normalizing the data.

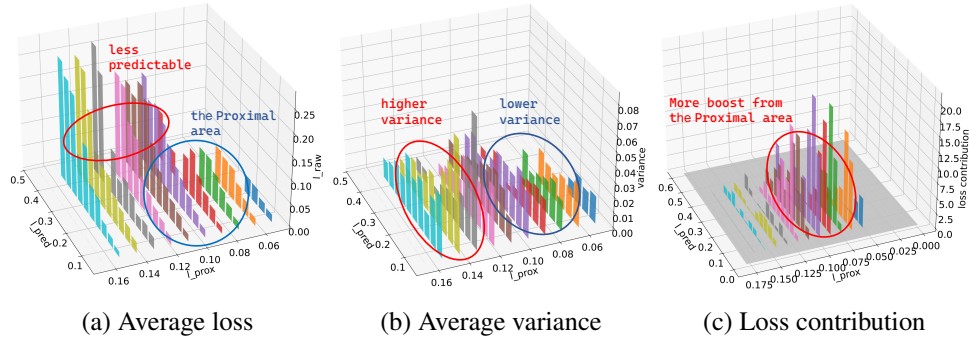

| (a) Average loss | (b) Average variance | (c) Loss contribution |
|---|---|---|

Figure 4: Visualization of the data distribution. **(a)** Each bar represents the average MSE loss measured on the test raw data w.r.t. proximity loss and prediction loss; **(b)** Each bar represents the average variance of the given series; **(c)** Each bar represents the total loss contribution modified by PTN, where the loss contribution for each bar equals (average loss × counts of samples).

**A Self-supervised Learner for Assessing Predictibility** When looking into the average loss distribution in the same setting as in Section 3.3, we can observe how data samples (points) are adaptively clustered into different distributions. Figure 4(a) shows that the average test MSE ("$l_{raw}$" on the z-axis) forms a clustering pattern for different data samples. The samples within the *proximal area* (blue, lower in proximity losses) have lower prediction losses on average, leaving the rest less predictable samples outside the area. This is also supported by the distribution of variance per sample (a data sample's variance is the variance of the series it belongs to), as shown in Figure 4(b). As variance is also a commonly considered metric w.r.t. predictability (lower variance for more predictable and higher variance for less predictable), we suggest that our method enables self-supervised estimates on the predictability and based on the estimates to constrain a proximal area for more predictable time series. We also analyze the distribution of performance boost, mea-

sured by loss contribution (average loss × number of samples) changed by PTN (with vs. without) in Figure 4. Performance boosts are mainly found in a limited range of proximity, supporting our assumption that effective data transformation exists within the proximal area close to the raw data. In essence, our PTN can categorize time series into predictable (low variance) and unpredictable (high variance) groups via a self-supervised manner, focusing on enhancing performance for the former.

**Transformation as Channel-adaptive** We analyze two representative channels from the ETTh2 dataset to illustrate Proximal Transformation. Figures 5(a) and (b) show the transformed series (in green) approximating the original (in blue). For Channel 1, the transformation evolves from oversmoothed to the original, while for Channel 5, it transitions from noisy to the original.

When comparing the distributions measured by the normalized values of both channels, it becomes evident that even after preprocessing with RevIN (Kim et al., 2021), the distributions of both channels are not aligned (see Figure 5(c)). We then apply our point-wise linear head to decode the output embeddings from the Convolution Encoder, resulting in the initially transformed data (in yellow). The distributions of both channels align (see Figure 5(d)), supporting the commonly adopted channel-independent strategy of sharing weights across channels. We provide more visualizations in Appendix A.7.2.

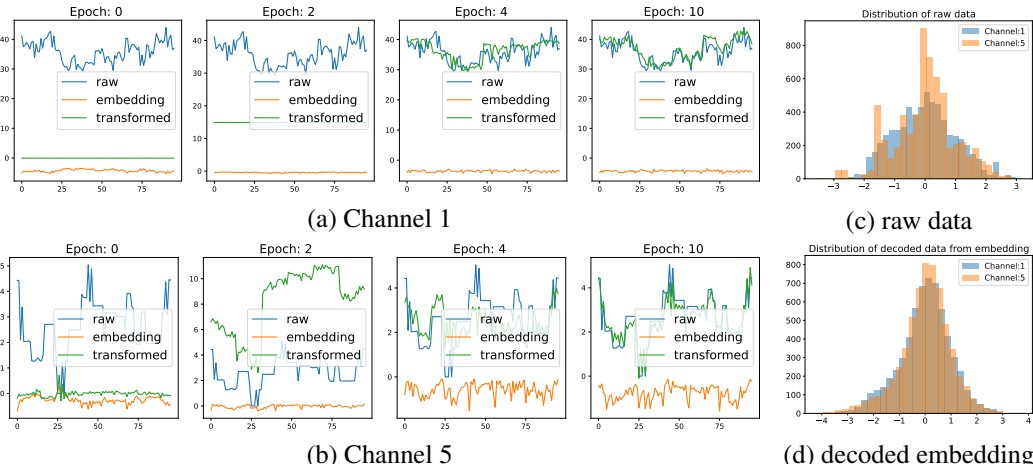

(a) Channel 1                                      (c) raw data

(b) Channel 5                              (d) decoded embedding

Figure 5: **(a) and (b)** Visualized raw and transformed series, respectively, in training PTN for Channels 1 and 5 in ETTh2 dataset; "raw" (blue) denotes the original series, "embedding" (yellow) the decoded series from the Convolution Encoder (Section 4.1); and transformed (green) the outputs of PTN. **(c)** The distributions of normalized raw series of Channels 1 and 5. **(d)** The distributions of decoded series of Channels 1 and 5.

## 5.3 TRANSFERABILITY OF THE TRANSFORMED DATA

Generalization capability is a commonly considered issue in machine learning. This topic is particularly important in TSF, as the diversity of datasets across various domains presents significant challenges in model design. Transfer learning adapts models to new datasets to enhance performance or reduce training costs. Our reformulating of finding a transformation of data offers a novel perspective on transfer learning. Specifically, our PTN design facilitates the combination of transformed data with various forecasting models. A simple, lightweight linear model can initially determine the transformation, which is then fixed and transferred to other forecasting models. Table 4 demonstrates the transfer learning results, showcasing the feasibility of our method. The data transfer protocol broadens conventional transfer learning by moving from generalizing a model for different datasets to its dual form of generalizing the data for different models.

## 5.4 ABLATION STUDY ON THE PTN DESIGNS

**Encoder** We propose using a CNN-based encoder (see Section 4.1) to generate an initial transformation of the raw time series. To evaluate its effectiveness, we analyze the following three variants: **1)**

Table 4: Comparison of generalization capabilities of our data transfer protocol in terms of MAE: 'A → B' indicates the raw data transformation is learned by Model A, and then fed to Model B.

| Model | RLINEAR → PATCHTST | | | | PATCHTST | | | | RLINEAR → iTRANSFORMER | | | | iTRANSFORMER | | | |
|---|---|---|---|---|---|---|---|---|---|---|---|---|---|---|---|---|
| Horizon | 96 | 192 | 336 | 720 | 96 | 192 | 336 | 720 | 96 | 192 | 336 | 720 | 96 | 192 | 336 | 720 |
| ETTh1 | **0.410** | **0.427** | **0.456** | 0.505 | 0.419 | 0.445 | 0.466 | **0.488** | 0.409 | **0.430** | **0.453** | 0.513 | **0.405** | 0.436 | 0.458 | **0.491** |
| ETTh2 | 0.349 | 0.402 | 0.443 | 0.526 | **0.348** | **0.400** | **0.433** | **0.446** | 0.400 | 0.403 | 0.450 | 0.619 | **0.349** | **0.400** | **0.432** | **0.445** |
| Electricity | **0.247** | **0.256** | **0.272** | **0.304** | 0.270 | 0.274 | 0.293 | 0.324 | **0.233** | **0.248** | **0.262** | **0.277** | 0.240 | 0.253 | 0.269 | 0.317 |
| Traffic | **0.274** | **0.277** | **0.283** | **0.302** | 0.290 | 0.290 | 0.300 | 0.320 | **0.242** | **0.253** | **0.264** | **0.281** | 0.268 | 0.276 | 0.283 | 0.302 |

Table 5: Experiments results of Ablation Study on our Convolution Encoder under different settings.

| | w. Trans. on $X$ | w. Trans on $Y$ | Embedding Type | ETTh1 MSE | ETTh1 MAE | ETTh2 MSE | ETTh2 MAE | ETTm1 MSE | ETTm1 MAE | ETTm2 MSE | ETTm2 MAE |
|---|---|---|---|---|---|---|---|---|---|---|---|
| PatchTrans | ✓ | ✓ | Patch | 0.444 | 0.436 | 0.471 | 0.452 | 0.399 | 0.390 | 0.279 | 0.329 |
| ConvPred | | ✓ | Conv | 0.490 | 0.463 | 0.529 | 0.485 | 0.389 | 0.393 | 0.343 | 0.372 |
| SingleTrans | ✓ | | Conv | 0.442 | 0.427 | 0.380 | 0.416 | 0.394 | 0.389 | 0.294 | 0.335 |
| ConvTrans (**ours**) | ✓ | ✓ | Conv | **0.437** | **0.427** | **0.366** | **0.391** | **0.386** | **0.390** | **0.272** | **0.319** |

*PatchTrans*: A common encoder alternative is patching. We perform experiments where the size of MoE is 1. Our design, named "ConvTrans", uses the CNN encoder, while the variant, "PatchTrans", uses the patching encoder. **2)** *ConvPred*: To test the encoder's effectiveness in a Transformer-based model, we replace the embedding module in PATCHTST with our CNN-based encoder. In fact, using the same decoder to predict is equivalent to approximating the transformation only on $Y$. **3)** *SingleTrans*: As analyzed in Section 3.1, we verify the need to transform both $X$ and $Y$. This variant only transforms $X$. The configurations of all ablation models are summarized in Table 5. Our findings show that our setting consistently outperforms the others.

**Decoder** Regarding decoder design, we primarily focus on two attention modules: Intra-Patch (IP) attention and Channel-wise (Ch) attention. The results are presented in Figure 6. As it demonstrates, combining both modules does not necessarily yield the best results across all datasets and backbones. However, datasets with more timestamps (e.g., ETTm1) benefit from IP attention, while datasets with more channels (e.g., Electricity) benefit from Ch attention. Therefore, we incorporate both modules to address these varying needs. This also frees us from excessive hyperparameter searches, allowing for more general considerations on different datasets and models.

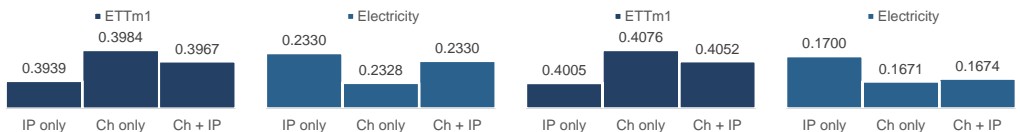

(a) PTN-RLI on ETTm1  (b) PTN-RLI on Electricity  (c) PTN-ITR on ETTm1  (d) PTN-ITR on Electricity

Figure 6: Ablation Study results of the Decoder for PTN-RLI and PTN-ITR (metric: MSE).

## 6 CONCLUSION AND FUTURE WORK

In this study, we approach time series forecasting (TSF) from a data-centric perspective by designing a co-optimization process that balances data transformation proximity and forecasting accuracy. Our design boosts the performance of a variety of TSF models and demonstrates the potential to make TSF more interpretable. For future work, it is relevant to align datasets from diverse sources using our data transformation method. Moreover, it remains open if PTNcan enhance performance in other time series tasks like anomaly detection, classification, and imputation. Further exploration into different backbone architectures is also interesting. We anticipate that a data-centric approach to generalizing datasets, tasks, and architectures could ideally pave the way for enhanced Time Series Foundation Models (Liang et al., 2024).

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

# A APPENDIX

## A.1 DISCUSSION ON PROPOSITION 1

We are going to briefly discuss Proposition 1 when $X_{train}$ and $X_{test}$ are identically distributed (same for $Y_{train}$ and $Y_{test}$, therefore we no longer use different subscripts to distinguish them). Specifically, we briefly discuss a very simple scenario that $X$ and $Y$ satisfies $Y = WX$, where $X \in \mathbb{R}^{L \times N}$ and $Y \in \mathbb{R}^{H \times N}$. And we have noised observations $Y = WX + \epsilon$ that parameters are trained on, where $\epsilon \in \mathbb{R}^{H \times N}$ and each elements in $\epsilon$ is sampled from $\mathcal{N}(\mu, \sigma)$

**Lemma 1** *Suppose we have a linear predictor $f(x; W_\theta) = W_\theta x$, the closed-form solution for $\min\limits_{W_\theta} ||W_\theta X - (Y + \epsilon)||_2$ is $W_\theta^* = (Y + \epsilon)X^\top(XX^\top)^{-1}$.*

*Proof.* $W_\theta^*$ can be solved by the following:

$$\frac{\mathrm{d}\left(W_\theta^* X - (Y + \epsilon)\right)^2}{W_\theta^*} = 0, \tag{6}$$

because

$$\frac{\mathrm{d}\left(W_\theta^* X - (Y + \epsilon)\right)^2}{W_\theta^*} = 2\left(W_\theta^* X - (Y + \epsilon)\right) X^\top. \tag{7}$$

We then solve Equation 6:

$$\begin{aligned}
2\left(W_\theta^* X - (Y + \epsilon)\right) X^\top &= 0 \\
\Leftrightarrow W_\theta^* X X^\top - (Y + \epsilon) X^\top &= 0 \\
\Leftrightarrow W_\theta^* &= (Y + \epsilon) X^\top \left(XX^\top\right)^{-1}
\end{aligned} \tag{8}$$

This follows the typical form of a linear regression problem, and this closed-form solution always exists because of its convex nature. $\square$

**Theorem 1** *We have $\tilde{W}_\theta = W$ that $\mathbb{E}||\tilde{W}_\theta X - (Y + \epsilon)||_2 < \mathbb{E}||W_\theta^* X - (Y + \epsilon)||_2$, when $XX^\top$ is slightly ill-conditioned that its condition number $\kappa(XX^\top) = ||XX^\top||_2 \cdot ||(XX^\top)^{-1}||_2 > 2$.*

*Proof.*

$$\begin{aligned}
\mathbb{E}||\tilde{W}_\theta X - (Y + \epsilon)||_2 &= \mathbb{E}||WX - (Y + \epsilon)||_2 \\
&= \mathbb{E}||Y - Y - \epsilon||_2 \\
&= \mathbb{E}||\epsilon||_2
\end{aligned} \tag{9}$$

$$\begin{aligned}
\mathbb{E}\left\|W_\theta^* - (Y + \epsilon)\right\|_2 &= \mathbb{E}\left\|(Y + \epsilon)X^\top(XX^\top)^{-1}X - (Y + \epsilon)\right\|_2 \\
&= \mathbb{E}\left\|(WX + \epsilon)X^\top(XX^\top)^{-1}X - (Y + \epsilon)\right\|_2 \\
&= \mathbb{E}\left\|W\left[XX^\top\left(XX^\top\right)^{-1}\right]X + \epsilon X^\top(XX^\top)^{-1}X - (Y + \epsilon)\right\|_2 \\
&= \mathbb{E}\left\|\epsilon\left[X^\top\left(XX^\top\right)^{-1}X - I\right]\right\|_2 \\
&= \mathbb{E}\frac{\left\|\epsilon\left[X^\top\left(XX^\top\right)^{-1}XX^\top - X^\top\right]\right\|_2}{\|X^\top\|_2} \\
&\geq \mathbb{E}\frac{\|\epsilon\|_2\|X^\top\|_2\left(\left\|\left(XX^\top\right)^{-1}XX^\top\right\|_2 - 1\right)}{\|X^\top\|_2} \\
&= \mathbb{E}\|\epsilon\|_2 \cdot \left(\left\|\left(XX^\top\right)^{-1}XX^\top\right\|_2 - 1\right)
\end{aligned} \tag{10}$$

so we have $\mathbb{E}\left\|\tilde{W}_\theta X - (Y + \epsilon)\right\|_2 < \mathbb{E}\|W_\theta^* X - (Y + \epsilon)\|_2$, when $\left\|XX^\top\right\|_2 \cdot \left\|\left(XX^\top\right)^{-1}\right\|_2 > 2$. $\square$

Table 6: Condition numbers for different datasets under the adopted look-back lengths.

|     | ETTh1 | ETTh2 | ETTm1 | ETTm2 | electricity | traffic | weather |
|-----|-------|-------|-------|-------|-------------|---------|---------|
| 96  | 6084.71 | 24703.02 | 25834.22 | 67423.96 | 4609.54 | 1710.74 | 2334938.58 |
| 192 | 12797.42 | 52742.00 | 52918.36 | 147632.44 | 10073.12 | 4646.02 | 4461250.74 |
| 336 | 24097.35 | 97711.27 | 94813.64 | 277571.90 | 19957.84 | 12035.87 | 7681729.05 |
| 720 | 76941.31 | 280377.34 | 217174.44 | 646904.38 | 53569.14 | 43433.10 | 16193817.02 |

* Note that the condition numbers are calculated by torch.linalg.inv and torch.linalg.matrix_norm. As the document suggests, the computation results are sensitive for non-square matrices. Therefore the figures here are for reference only and to provide some insights on future research such as why or why not simple linear models can outperform Transformer-based models on certain datasets.

This simple example is meaningful for a variety of linear models covering temporal and frequency domains, as they are equivalent to the closed-formed solution in **Lemma 1** Toner & Darlow (2024). In real scenarios, the raw data is often ill-conditioned as discussed in Toner & Darlow (2024) (shown in Table 6). By the above Theorem, we demonstrate that under very trivial conditions, Proposition 1 often holds when using linear models, even if the train sets and test sets are identically distributed.

## A.2 RELATED WORK REVISIT

In this part, we will introduce some popularly adopted methods for TSF, in which our baseline models are included.

**Channel-independent Transformers**  Time Series Transformers (TST) (Zhou et al., 2021; Wu et al., 2021; Zhou et al., 2022) have recently led to significant progress in the TSF problem, demonstrating convincing superiority over traditional methods and convolution-based models.

Inspired by Zeng et al. (2023), Nie et al. (2023) incorporates the *Channel Independent* (CI) design (sharing weights across all channels) to introduce PATCHTST, a new state-of-the-art (SOTA) model that significantly benefits from CI and the patching operation. More works follow this practice and achieve excellent performance in TSF (Das et al., 2024; Liu et al., 2024a; Zhang et al., 2024).

**Channel-wise Transformers**  Building on this, more recent research has focused on designing Transformers capable of capturing channel dependencies inherent in multivariate time series data. Notable examples include CROSSFORMER (Zhang & Yan, 2023), iTRANSFORMER (Liu et al., 2024b), DSFORMER (Yu et al., 2023), and CARD (Wang et al., 2024b). These models shed light on exploiting inter-series relationships to improve forecasting accuracy. Furthermore, Ilbert et al. (2024) propose that Transformers are inherently unstable when trained on time series datasets due to sharp gradient curvature, and suggest a sharpness-aware optimizer to mitigate such issues. Nonetheless, channel-wise Transformers still suffer from overfitting on small datasets.

**Linear Models**  In contrast to the quadratic complexity of Transformers, lightweight linear models have emerged as competitive alternatives offering simplicity and efficiency. RLINEAR (Li et al., 2023) verifies that a vanilla single-linear model, when combined with a widely-adopted normalization method (Kim et al., 2021), can achieve near SOTA performance in TSF. Further research (Zhao & Shen, 2024) on channel dependencies within linear and MLP-based models has yielded performance improvements over previous CI approaches. Moreover, models (Xu et al., 2024; Yi et al., 2024) that directly learn linear regression or MLP-based models on complex frequency features have achieved remarkable performances. Through theoretical analysis of these linear models, Toner & Darlow (2024) conclude that normalization-enhanced (Li et al., 2023), decomposition-based (Zeng et al., 2023) and frequency-domain linear (Xu et al., 2024) models are essentially equivalent to vanilla linear regression. Most recently, SPARSETSF (Lin et al., 2024), a SOTA lightweight model, incorporates 1D convolutions as down-sampling modules and learns linear parameters on the down-sampled values of the original series.

## A.3 Experimetal Details

### A.3.1 Datasets

We conduct experiments on 11 real-world datasets to evaluate the performance of the proposed PTN. (1) ETT (Zhou et al., 2021) contains 7 factors of electricity transformers from July 2016 to July 2018. The ETTh1 and ETTh2 are subsets that are recorded every hour, and ETTm1 and ETTm2 are recorded every 15 minutes. (2) Electricity (Wu et al., 2021) records the hourly electricity consumption data of 321 clients. (3) Traffic (Wu et al., 2021) collects hourly road occupancy rates measured by 862 sensors of San Francisco Bay area freeways from January 2015 to December 2016. (4) Weather (Wu et al., 2021) includes 21 meteorological factors collected every 10 minutes from the Weather Station of the Max Planck Biogeochemistry Institute in 2020. (5) PeMS contains the public traffic network data in California collected by 5-minute windows. We use the same four public subsets (PeMS03, PeMS04, PeMS07, PeMS08) adopted in iTransformer (Liu et al., 2024b). For ETT datasets, we divide datasets by ratio $\{0.6, 0.2, 0.2\}$ into train set, validation set, and test set. For Electricity, Traffic, Weather and PeMS datasets, we divide by ratio $\{0.7, 0.1, 0.2\}$ in the same setting of TimesNet (Wu et al., 2023) and many previous works. All datasets are scaled by the mean and variance of train sets, a common practice in TSF (Wu et al., 2023).

Table 7: Statistics of evaluation datasets.

| Datasets | ETTh1 | ETTh2 | ETTm1 | ETTm2 | Electricity | Traffic | Weather | PeMS03 | PeMS04 | PeMS07 | PeMS08 |
|---|---|---|---|---|---|---|---|---|---|---|---|
| # of TS Variates | 7 | 7 | 7 | 7 | 321 | 862 | 21 | 358 | 307 | 883 | 170 |
| TS Length | 17420 | 17420 | 69680 | 69680 | 26304 | 17544 | 52696 | 26209 | 16992 | 28224 | 17856 |

### A.3.2 Baselines

For horizontal comparisons, we include RLINEAR, DLINEAR, SPARSETSF, iTRANSFORMER, PATCHTST, FEDFORMER, and TIMESNET, covering linear, transformed-based, convolution-based and frequecy-domain methods. (1) RLINEAR (Li et al., 2023), a linear model with Reversible Instance Normalization (Kim et al., 2021); (2) DLINEAR (Zeng et al., 2023), a linear model on the decomposed Trend term and Seasonal term; (3) SPARSETSF (Lin et al., 2024). A linear model using sparse technique; (4) iTRANSFORMER (Liu et al., 2024b), a Transformer-based model the compute attention score on the inverted series (along the channel dimension); (5)PATCHTST (Nie et al., 2023), a Channel-Independent Transformer-based model that uses patching to tokenize; (6) FEDFORMER, a Transformer-based model utilizing frequency-domain information. The Table 7 shows the detailed statistics of 7 datasets. It should be pointed out that the scale of a time series dataset is determined by the number of variates and dataset length combined. Therefore a more fair comparison should cover both two factors. In our experiment, we observe that some baseline models like (Liu et al., 2024b) can benefit greatly from a larger number of variates, while some may prefer longer datasets (Nie et al., 2023) or longer look-back lengths (Lin et al., 2024).

## A.4 Implementation Details

### A.4.1 Hyperparameters and Settings

All experiments and methods are implemented in Python and PyTorch (Paszke et al., 2019) and conducted on two Nvidia RTX A5000 ada generation GPUs (32GB) and two Nvidia RTX A6000 GPUs (48GB). We use ADAM (Kingma, 2014) and the learning rate initialized as 0.0003 for all settings. Instead of setting an exceptionally small epoch for all experiments as adopted in previous works (Liu et al., 2024b; Lin et al., 2024), we use early stopping on the MSE metric of the validation set with patience set as 10 epochs.

The hyperparameters of convolution encoder will be set as fixed as will be explained in Section A.4.5. We the size of hidden dimension of attention-based decoder as 32 and use two attention layers for both intra-patch and channel-wise modules. The patch length also referred as the period length in SparseTSF Lin et al. (2024) is set to 4 to support an enffient implmentation. The total number of parameters are 26285, which is a relatively light amount of parameters when counting

static memory, i.e. the memory used to store parameters. For more practical scenes when it comes to time and memory complexity, we are going to analyze in the following.

### A.4.2 OVERALL PIPELINES OF PTN IN DIFFERENT WORKING MODES

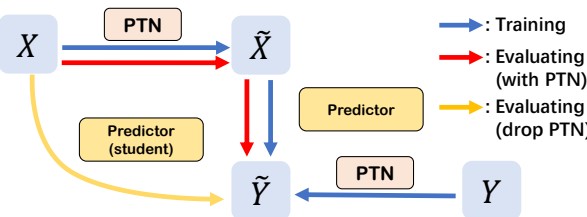

Figure 7: The overall pipelines for PTN in different working modes.

As illusrated in Figure 3, the proposed Proximal Transformation Network (PTN) consists of three distinct pipelines: one for training, one for evaluation with PTN, and one for evaluation without PTN.

During the training phase, the pipeline involves two transformation processes applied to both $X$ and $Y$ (see blue arrows in Figure 3), followed by the forecasting process, which is parameterized by an arbitrary predictor.

For evaluation (inference), there are two options available. The first option (see the red arrow) is the default evaluation, applicable to any plugin module, where the backbone model (the predictor in this case) is utilized. The second option (see the yellow arrow) provides a simplified approach for evaluation or inference. In this case, the raw data $X$ is fed directly into a *distilled* student model, which has the same architecture and produces outputs trained using $\tilde{Y}$ as labels. Further details regarding the student model are provided in Appendix A.4.4.

### A.4.3 COMPLEXITY ANALYSIS OF PTN

The proposed PTN serves as a plugin that can enhance a backbone's performance at the cost of additional computation and memory usage. Hence, we proceed to analyze the computational complexity of PTN, bridging some of the gaps between theoretical complexity from past studies and our empirical findings. For a brief conclusion, our complexity analysis offers guidance on implementing PTN efficiently for practical usage. By limiting the size of parallel computations within a batch, we can reduce the relative multiplier of time and memory costs from a number of times to a factor of less than one during training and inference.

To be specific, we first identify the actual batch size as a crucial factor in complexity analysis and extend the analysis from previous studies to better align with the empirical results (see P1 below). Second, by reducing actual batch size, we propose a more efficient implementation for the combinational uses of two orthogonal attentions (core components in PTN). Under certain conditions, this method can achieve reasonable speed-up at the cost of memory usage (see P2). Third, we verify the effectiveness of the variate sampling strategy, which is also supported by our analysis (see P3). Both methods here do not incur performance drop with improvement in efficiency.

**P1. Complexities of different Attentions**. A previous study by Liu et al. (2024b) reveals that the channel-wise and inter-patch attentions have different complexities based on dataset features. Compared to inter-patch attention, channel-wise attention uses more resources on datasets with more variables and less on those with fewer variables. This difference is due to the dimension along which attention is calculated. The complexity of attention is $\mathcal{O}(N^2)$, where $N$ represents the number of tokens. For channel-wise attention, $N$ is the number of variables, while for inter-patch attention, $N$ is the number of patches. These complexities are represented as $\mathcal{O}(C^2)$ for channel-wise and $\mathcal{O}(n^2)$ for inter-patch, where $C$ is the number of variables and $n$ is the number of patches (calculated as $L/l_p$, with $l_p$ being the patch length). Intra-patch attention has a constant complexity of $\mathcal{O}(l_p^2)$, meaning it doesn't scale with the series length or horizon.

However, despite not computing along the variable dimension, the times of inter-patch and intra-patch attention computation increase nearly linearly with more variables. Due to channel-independence strategy Zeng et al. (2023), the number of variates will multiply the batch size, resulting in a larger number of parallel computations. Since GPU parallel capacity is limited, a larger batch size does not necessarily speed up computation. What is more, the increased batch size does not lower the number of batches to be computed in a dataset. Therefore we should take actual batch size into consideration for complexity. For large $C$, inter-patch attention complexity is $\mathcal{O}(C \times n^2)$. As shown in Figure 10 and Figure 11, datasets with many variables (like the Traffic dataset with 862 variables) result in faster training and inference speeds for models like PATCHTST and DLINEAR compared to datasets with fewer variables (like the Weather dataset with 21 variables). This leads us to adjust the empirical computation complexity of intra-patch attention. We compute intra-patch attention for $n \times C = L/l_p \times C$ times, resulting in a complexity of $\mathcal{O}(l_p \cdot LC)$. Similarly, channel-wise attention complexity is $\mathcal{O}(L \cdot C^2)$. By adding the outputs of both attentions, the total complexity becomes $\mathcal{O}((l_p + C) \cdot LC)$.

**P2. Mask Fusion**. As the analysis may suggest, constraining the actual batch size by computing longer sequences of attention may increase efficiency in cases. Indeed, we have developed an alternative approach to implement our attention mechanism, which can reduce computational complexity in certain scenarios. As discussed, the attention components $(Q, K, V)$ have a shape of $[B, N, D]$, where variations in sequence length $N$ and the batch size $B$ affect practical complexity with given GPU parallel capacity. In this sense, increasing the sequence length can enhance efficiency in some cases by having a smaller number of $B$.

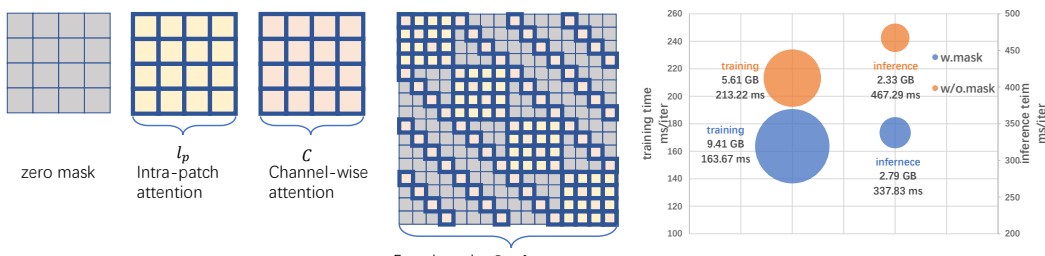

Figure 8: Fuse two attention masks in one

Figure 9: Empirical results of fusing masks with fixed 64 variates

Figure 8 demonstrates that by fusing the attention masks, we can compute intra-patch and channel-wise attention simultaneously. This results in a fused attention complexity of $\mathcal{O}(l_p^2 C^2)$. When $l_p^2 C < (l_p + C)L$, this fused approach is more efficient than the standard implementation. In practice, we use a small patch length of $l_p = 4$. For empirical analysis, we set a batch size of 128, patch length of 4, and a number of variables as 64 to evaluate the effectiveness of mask fusion. As shown in Figure 9, time consumption is reduced by 23.34% during training and 27.70% during inference, though it requires more memory.

**P3. Variate Sampling**. According to (Liu et al., 2024b), even for channel-dependent methods, sampling variables from different channels does not bring performance degrades while significantly improving efficiency. As analyzed, this technique should reduce complexity related to the number of channels and apply to channel-independent methods as well. We also compare memory and time complexity with this sampling strategy, using a sampling size of 64. The results are presented in Table 8 and Table 9.

For a more comprehensive empirical study, we conducted complexity experiments on Traffic and Weather datasets, in line with previous work like SPARSETSF and iTRANSFORMER. We also included DLINEAR to represent linear models, which should have similar complexities. The results are illustrated in Figure 10 and Figure 11.

Additionally, inference and training cost increases may differ across different backbones. For instance, training speed might decrease more for DLINEAR and iTRANSFORMER but maintain a relatively low decrease for PATCHTST, potentially reducing time complexity due to efficient attention mask implementation. The efficiency-friendly methods here do not sacrifice performance. An al-

Table 8: Training complexity increases under different settings on Traffic and Weather datasets. For the sampling strategy, we compute the training time of only the sampled variates. The time complexity is measured by "ms/iter" for the time consumed for each iteration. The memory complexity is measured by "GB" for GPU memory consumed. The additional cost is measured by "inc%" for the relative increase in time or memory compared with backbone models.

| Models | | DLINEAR | | | | iTRANSFORMER | | | | PATCHTST | | | |
|---|---|---|---|---|---|---|---|---|---|---|---|---|---|
| metric | | time | | memory | | time | | memory | | time | | memory | |
| | | ms/iter | inc% | GB | inc% | ms/iter | inc% | GB | inc% | ms/iter | inc% | GB | inc% |
| Traffic | w./o. sampling | 150.105 | 710.45% | 6.830 | 563.58% | 149.201 | 82.51% | 7.928 | 62.57% | 79.701 | 31.56% | 3.330 | 33.38% |
| | w. sampling | 10.775 | 51.92% | 0.042 | 3.83% | 9.383 | 36.56% | 0.033 | 2.48% | 4.779 | 15.74% | -0.139 | -8.64% |
| Weather | w./o. sampling | 9.452 | 86.11% | 0.070 | 8.66% | 4.301 | 30.15% | 0.039 | 3.48% | 4.429 | 24.87% | -0.137 | -9.84% |
| | w. sampling | 2.449 | 28.30% | 0.035 | 8.39% | 1.941 | 18.40% | 0.023 | 3.90% | -0.690 | -5.28% | -0.008 | -1.02% |

Table 9: Inference complexity increases under different settings on Traffic and Weather datasets. For the sampling strategy, we compute the inference time of evaluating all variates. The time complexity is measured by "ms/iter" for the time consumed for each iteration. The memory complexity is measured by "GB" for GPU memory consumed. The additional cost is measured by "inc%" for the relative increase in time or memory compared with backbone models.

| Models | | PTN-DLI | | | | PTN-ITR | | | | PTN-PAT | | | |
|---|---|---|---|---|---|---|---|---|---|---|---|---|---|
| metric | | time | | memory | | time | | memory | | time | | memory | |
| | | ms/iter | inc% | GB | inc% | ms/iter | inc% | GB | inc% | ms/iter | inc% | GB | inc% |
| Traffic | w./o. sampling | 41.525 | 237.19% | 1.705 | 301.55% | 50.566 | 81.66% | -0.223 | -4.13% | 20.872 | 21.35% | -0.039 | 3.03% |
| | w. sampling | 29.252 | 133.74% | 0.002 | 0.31% | 29.260 | 88.13% | 0.039 | 5.79% | -4.473 | -5.94% | 0.004 | 0.54% |
| Weather | w./o. sampling | 2.047 | 23.54% | 0.008 | 1.98% | 1.963 | 20.31% | 0.055 | 11.59% | 1.101 | 11.13% | 0.021 | 4.55% |
| | w. sampling | 2.449 | 28.30% | 0.035 | 8.39% | 1.941 | 18.40% | 0.023 | 3.90% | -0.690 | -5.28% | -0.008 | -1.02% |

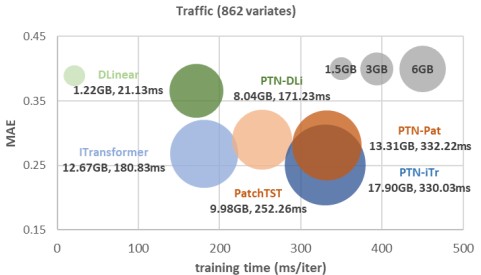

(a) Training complexity for Traffic dataset with all variates

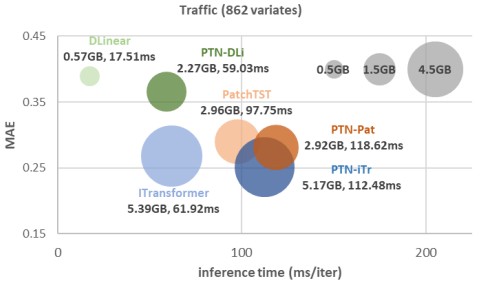

(b) Inference complexity for Traffic dataset with all variates

(c) Training complexity for Traffic dataset with variate sampling

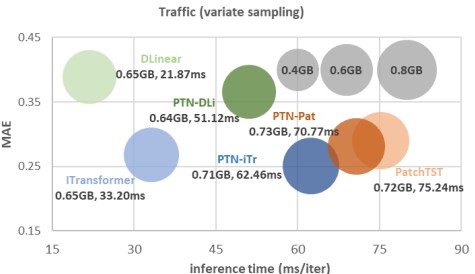

(d) Inference complexity for Traffic dataset with variate sampling

Figure 10: Visualization of computation and memory complexity on Traffic dataset

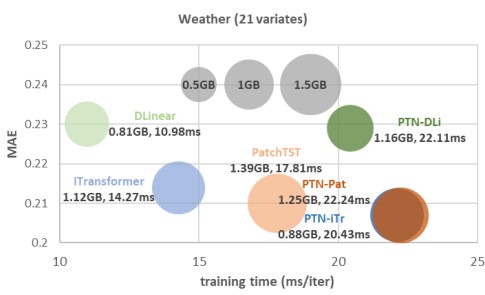

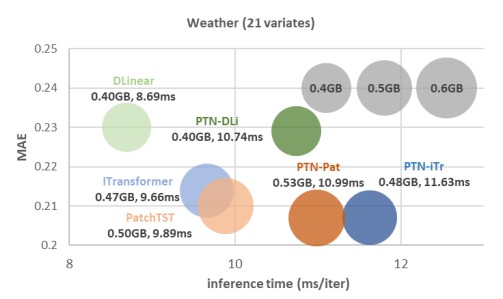

(a) Training complexity for Weather dataset with all variates

(b) Inference complexity for Weather dataset with all variates

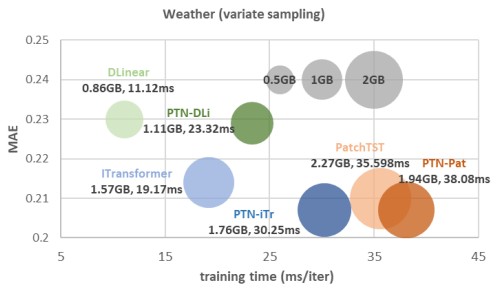

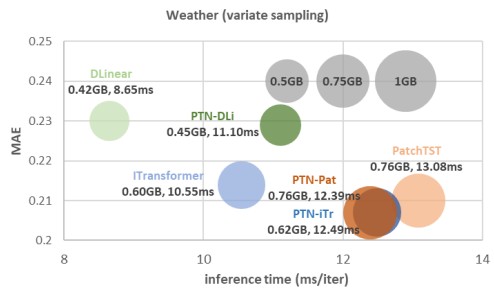

(c) Training complexity for Weather dataset with variate sampling

(d) Inference complexity for Weather dataset with variate sampling

Figure 11: Visualization of computation and memory complexity on Weather dataset

ternative option, detailed in Appendix A.4.4, incurs no additional inference costs with performance compromises, allowing for a flexible trade-off between efficiency and effectiveness.

### A.4.4 EFFICIENT STUDENT MODEL BY TRAIN-TIME DISTILLATION

The training cost for non-linear attention-based models can be reduced by a data transfer protocol as described in Section 5.3. The transfer utilizes the transformation learned on a linear backbone to train a non-linear model from scratch, which improves training efficiency at a minor compromise of performance. We have a more direct and burden-free version for inference as well, ss shown in Figure 7 and briefly explained in the previous section, The core idea is to train a student model that directly uses the raw input $X$ to predict the transformed output $\tilde{Y}$. This practice is an intermediate version between a standard PTN and the data transfer version. Unlike the data transfer protocol in Section 5.3, the student model learns the mapping from $X$ to $\tilde{Y}$ during training, which means it requires more samples than just the converged $\tilde{Y}$ after training PTN. When training a standard PTN, the backbone predictor uses the transformed series as inputs and labels, and updates computed by the $l_pred$ will also back-propagate the gradients of the PTN parameters. Whereas when training with a student predictor, we use straight-through method (Van Den Oord et al., 2017) to omit the calculation of PTN gradients (i.e. use "tensor.detach()" in pytorch). Thereafter, we only added the cost of another backbone model and can achieve similar improvements without any additional cost increase during the inference phase.

### A.4.5 CONVOLUTION ENCODER

**Transpose and Unfold**  The implementation details regarding Convolution Encoder involve two operations that are designed for the following two benefits. (1) The convolution outputs can be concatenated in the same length of embedding so that features from different frequencies can be ideally fused into one uniform embedding. (2) The features can be evenly arranged along the temporal dimension so that each embedding in the sequence can have the same large Receptive Field. There-

Table 10: Full results for performance comparisons between backbone models and student models with train-time distillation by PTN.

| Models | | DLINEAR | | +PTN(stu) | | PATCHTST | | +PTN(stu) | | iTRANSFORMER | | +PTN(stu) | |
|---|---|---|---|---|---|---|---|---|---|---|---|---|---|
| Metric | | MSE | MAE | MSE | MAE | MSE | MAE | MSE | MAE | MSE | MAE | MSE | MAE |
| ETTh1 | 96 | 0.386 | 0.395 | **0.384** | **0.391** | 0.414 | 0.419 | **0.396** | **0.411** | **0.386** | **0.405** | 0.401 | 0.409 |
| | 192 | 0.437 | 0.424 | **0.436** | **0.422** | 0.460 | 0.445 | **0.455** | 0.447 | **0.441** | **0.436** | 0.456 | 0.449 |
| | 336 | **0.479** | 0.446 | 0.480 | **0.444** | 0.501 | 0.466 | **0.496** | **0.473** | **0.487** | **0.458** | 0.540 | 0.504 |
| | 720 | 0.481 | 0.470 | **0.475** | **0.464** | 0.500 | **0.488** | 0.492 | 0.490 | **0.503** | **0.491** | 0.594 | 0.554 |
| | Avg. | 0.446 | 0.434 | **0.444** | **0.430** | 0.469 | **0.455** | 0.460 | 0.456 | **0.454** | **0.448** | 0.498 | 0.479 |
| ETTh2 | 96 | 0.288 | 0.338 | **0.284** | **0.335** | 0.302 | 0.348 | **0.296** | **0.347** | **0.297** | 0.349 | 0.302 | **0.349** |
| | 192 | 0.374 | 0.390 | **0.370** | **0.387** | 0.388 | 0.400 | **0.377** | **0.396** | **0.380** | **0.400** | 0.392 | 0.410 |
| | 336 | 0.415 | 0.426 | **0.414** | **0.424** | 0.426 | 0.433 | **0.423** | **0.437** | **0.428** | **0.432** | 0.451 | 0.449 |
| | 720 | 0.420 | 0.440 | **0.416** | **0.436** | 0.431 | **0.446** | 0.447 | 0.459 | **0.427** | **0.445** | 0.603 | 0.550 |
| | Avg. | 0.374 | 0.399 | **0.371** | **0.396** | 0.387 | **0.407** | 0.386 | 0.410 | **0.383** | **0.407** | 0.437 | 0.440 |
| Electricity | 96 | 0.201 | 0.281 | 0.206 | **0.278** | 0.181 | 0.270 | **0.174** | **0.258** | 0.148 | 0.240 | **0.143** | **0.234** |
| | 192 | 0.201 | 0.283 | 0.205 | **0.281** | 0.188 | 0.274 | **0.184** | **0.268** | 0.162 | 0.253 | **0.154** | **0.247** |
| | 336 | 0.215 | 0.298 | 0.219 | **0.296** | 0.204 | 0.293 | **0.200** | **0.284** | 0.178 | 0.269 | **0.169** | **0.263** |
| | 720 | 0.257 | 0.331 | 0.260 | **0.328** | 0.246 | 0.324 | **0.239** | **0.316** | 0.225 | 0.317 | **0.209** | **0.300** |
| | Avg. | 0.219 | 0.298 | 0.222 | **0.296** | 0.205 | 0.290 | **0.199** | **0.281** | 0.178 | 0.270 | **0.169** | **0.261** |
| Traffic | 96 | **0.649** | 0.389 | 0.664 | **0.384** | **0.462** | 0.290 | 0.474 | **0.277** | **0.395** | 0.268 | 0.406 | **0.255** |
| | 192 | **0.601** | 0.366 | 0.615 | **0.354** | **0.466** | 0.290 | 0.478 | **0.279** | **0.417** | 0.276 | 0.436 | **0.276** |
| | 336 | **0.609** | 0.369 | 0.621 | **0.357** | **0.482** | 0.300 | 0.487 | **0.287** | **0.433** | 0.283 | 0.448 | **0.275** |
| | 720 | **0.647** | 0.387 | 0.657 | **0.380** | **0.514** | 0.320 | 0.525 | **0.303** | **0.467** | 0.302 | 0.476 | **0.291** |
| | Avg. | **0.627** | 0.378 | 0.639 | **0.369** | **0.481** | 0.300 | 0.491 | **0.286** | **0.428** | 0.282 | 0.442 | **0.274** |

fore, we introduce the two-step operation of Transpose and Unfold, which brings the two benefits together. Specifically, we set kernel size = 3, stride = 2, paddig = 1, and the number of kernels doubled for each next layer. In this way, as shown in Figure 12, we can ensure the number of features is invariant for different layers, and only the shape is changed. Now we can fuse the outputs from different convolution layers together by flattening/unfolding the features to the original shape of $L \times 1$. Again, considering the effectiveness of point-wise linear head 4.1, we want the concatenated features to be near-equally arranged along the temporal dimension to maintain the sequential relationship in the embedding. So we transpose the features first and unfold them afterwards. This practice can ensure a Receptive Field of $(2^{l+1} - 1)$ wide for each embedding, where $l$ is the total number of convolution layers.

**Effective Receptive Field of Convolution Encoder**  As what was proposed by (Luo & Wang, 2024), the Effective Receptive Field (ERF) is a reasonable consideration for designing convolution-based architecture. We also input the impulse function into our Convolution Encoder to visualize the ERF, as shown in Figure 13. It has shown that without using an extra large convolution kernel, our proposed method can utilize a near-global ERF by combining outputs from different layers, with different frequency patterns.

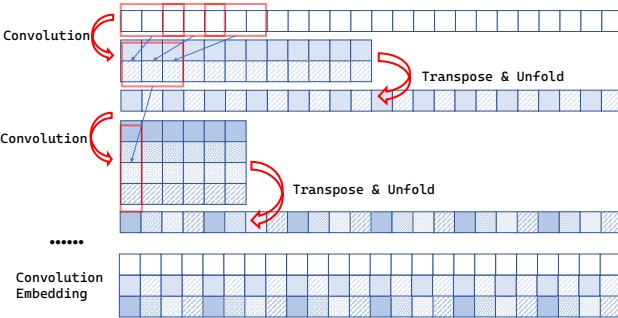

Figure 12: The illustration of transpose and unfold operation in the Convolution Encoder

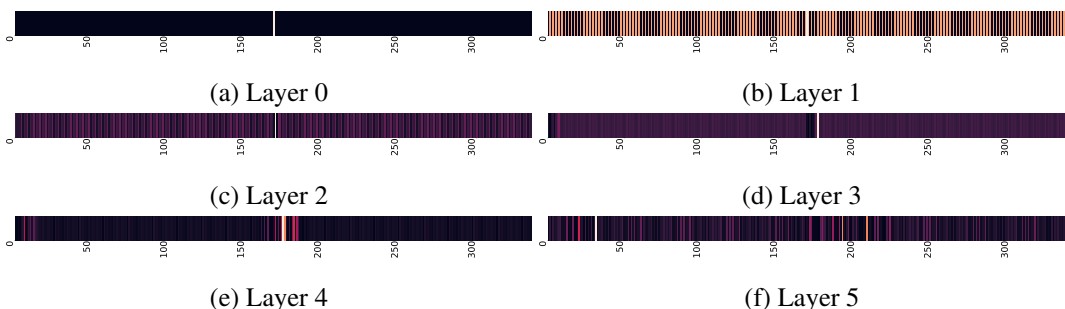

(a) Layer 0        (b) Layer 1

(c) Layer 2        (d) Layer 3

(e) Layer 4        (f) Layer 5

Figure 13: Effective Receptive Field (ERF) of our proposed Convolution Encoder.

### A.4.6 POINT-WISE LINEAR HEAD

As mentioned in Section 4.1, we employ a point-wise linear head as a parameter-sharing module. We demonstrate the three different parameterizations for the linear head in common TSF models in Figure 14. PATCHTST(Nie et al., 2023) adopts a patch-dependent linear head (shown in Figure 14(1)) that first flattens all features in patches and uses an extra large weight matrix with a shape of $Ld \times d$, where L is sequence length and d the dimension for latent embedding. Lee et al. (2023) proposes an intriguing idea that some tasks involve patch-dependent design as a necessity such as forecasting, while other tasks like contrastive learning might prefer patch-independent design (shown in Figure 14(b)). Our proximity approach process resembles a reconstruction process, therefore we do not exploit the patch correlations in our task and can even extend this independence to point-wise scope (shown in Figure 14(c)). This approach is feasible only when each point-wise embedding is rich in information, as supported by our Convolution Encoder with a near-global ERF for each point.

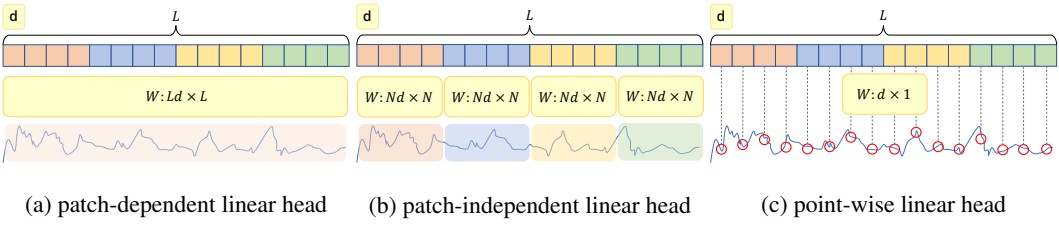

(a) patch-dependent linear head    (b) patch-independent linear head    (c) point-wise linear head

Figure 14: Three types of Linear Head.

## A.5 STUDY ON MIXTURE-OF-EXPERTS

### A.5.1 SCALING PARAMETERS

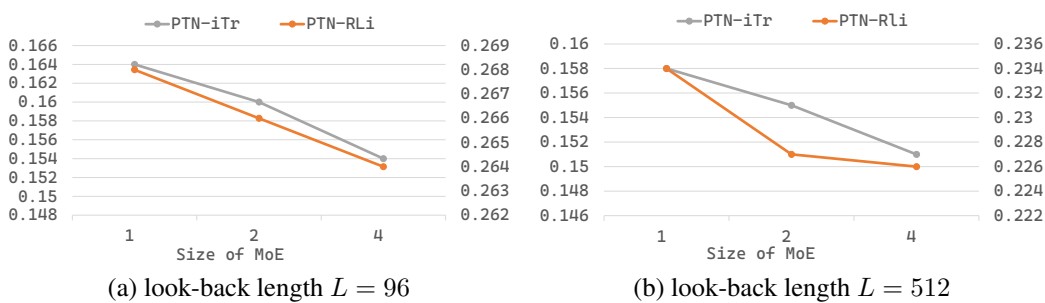

(a) look-back length $L = 96$        (b) look-back length $L = 512$

Figure 15: Performance boost by scaling the MoE.

Like any Mix-of-Experts model, increasing the size of MoE can help generate better results. We conduct such a hyperparameter study of two versions of our PTN model on the Weather dataset. The results are reported in Fig 15.

### A.5.2 REVISITING TIME SERIES MIXTURE-OF-EXPERTS: A DECOMPOSITION VIEW

Time series MoE has been recently proposed as an effective architecture in scaling parameters for time series models (Shi et al., 2024; Liu et al., 2023; Ni et al., 2024). Here, we conduct simple attempts to extend our MoE model to a larger number of 12, without subseries routing (i.e., each Expert transforms the whole series). This practice can be computation-intensive and beyond our capacity for extensive research. However, when we train the larger-in-scale MoE on par with its simplified counterpart (with subseries routing) on ETTh1, the visualization for each Expert shows patterns as a decomposition of time series, as shown in Figure 16.

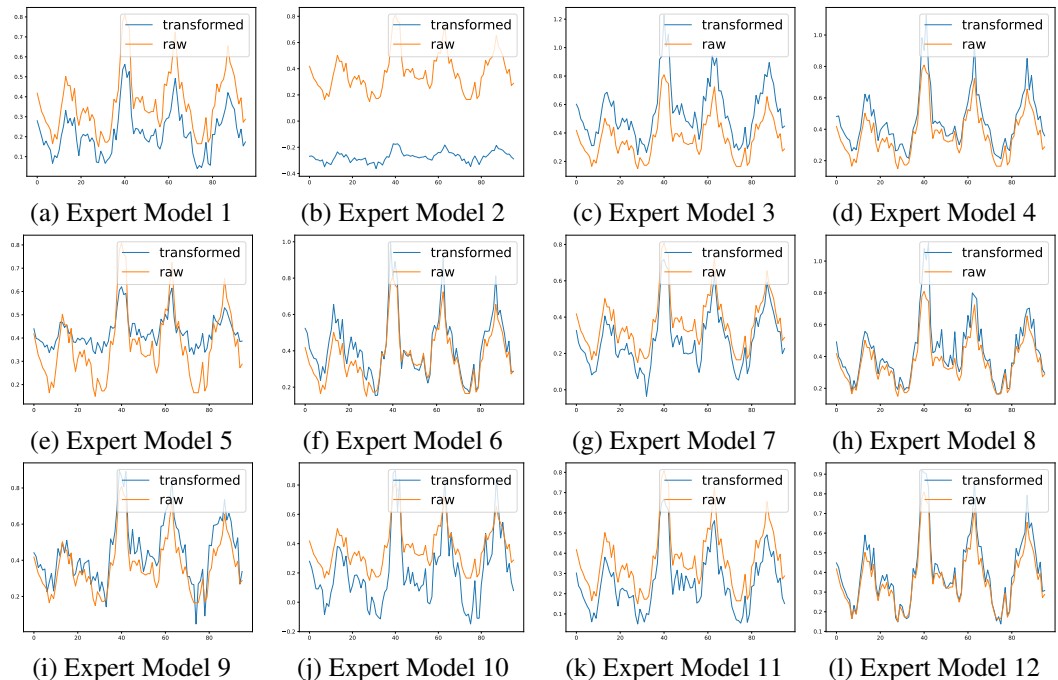

| (a) Expert Model 1 | (b) Expert Model 2 | (c) Expert Model 3 | (d) Expert Model 4 |
| (e) Expert Model 5 | (f) Expert Model 6 | (g) Expert Model 7 | (h) Expert Model 8 |
| (i) Expert Model 9 | (j) Expert Model 10 | (k) Expert Model 11 | (l) Expert Model 12 |

Figure 16: Transformed series from different Expert Models under full-transform setting.

### A.6 MORE RESULTS OF EXPERIMENTS

Table 11 and Table 12 show the full results under the look-back length $L = 96$ and $L = 512$ settings respectively. Tab 13 shows full results of a comparison study between popular backbones and them integrated with PTN-RLI.

Table 11: Multivariate long-term forecasting results of PTN-RLI (i.e. PTN with RLINEAR backbone) and PTN-ITR (i.e. PTN with ITRANSFORMER backbone). We use look-back length fixed as **96** and prediction lengths $T \in \{96, 192, 336, 720\}$ for all datasets. The best results are in **bold** and the second best is underlined.

| Models | | PTN-RLI | | PTN-ITR | | ITRANSFORMER | | PATCHTST | | TIMESNET | | SPARSETSF | | DLINEAR | | FEDFORMER | |
|---|---|---|---|---|---|---|---|---|---|---|---|---|---|---|---|---|---|
| Metric | | MSE | MAE | MSE | MAE | MSE | MAE | MSE | MAE | MSE | MAE | MSE | MAE | MSE | MAE | MSE | MAE |
| ETTh1 | 96 | **0.372** | **0.385** | 0.389 | 0.407 | 0.386 | 0.405 | 0.414 | 0.419 | 0.384 | 0.402 | 0.391 | 0.389 | 0.386 | 0.395 | 0.376 | 0.419 |
| | 192 | 0.431 | **0.418** | 0.432 | 0.432 | 0.441 | 0.436 | 0.460 | 0.445 | 0.436 | 0.429 | 0.441 | 0.419 | 0.437 | 0.424 | **0.420** | 0.448 |
| | 336 | 0.463 | **0.438** | 0.460 | 0.443 | 0.487 | 0.458 | 0.501 | 0.466 | 0.491 | 0.469 | 0.481 | 0.439 | 0.479 | 0.446 | **0.459** | 0.465 |
| | 720 | **0.464** | 0.460 | 0.532 | 0.523 | 0.503 | 0.491 | 0.500 | 0.488 | 0.521 | 0.500 | 0.467 | **0.453** | 0.481 | 0.470 | 0.506 | 0.507 |
| | Avg. | **0.437** | **0.427** | 0.453 | 0.451 | 0.454 | 0.448 | 0.469 | 0.455 | 0.458 | 0.450 | 0.445 | 0.425 | 0.446 | 0.434 | 0.440 | 0.460 |
| ETTh2 | 96 | **0.281** | **0.330** | 0.323 | 0.367 | 0.297 | 0.349 | 0.302 | 0.348 | 0.340 | 0.374 | 0.310 | 0.338 | 0.288 | 0.338 | 0.358 | 0.397 |
| | 192 | **0.361** | **0.380** | 0.377 | 0.399 | 0.380 | 0.400 | 0.388 | 0.400 | 0.402 | 0.414 | 0.391 | 0.394 | 0.374 | 0.390 | 0.429 | 0.439 |
| | 336 | **0.409** | **0.418** | 0.453 | 0.450 | 0.428 | 0.432 | 0.426 | 0.433 | 0.452 | 0.452 | 0.424 | 0.428 | 0.415 | 0.426 | 0.496 | 0.487 |
| | 720 | **0.413** | **0.434** | 0.610 | 0.552 | 0.427 | 0.445 | 0.431 | 0.446 | 0.462 | 0.468 | 0.421 | 0.437 | 0.420 | 0.440 | 0.463 | 0.474 |
| | Avg. | **0.366** | **0.391** | 0.441 | 0.442 | 0.383 | 0.407 | 0.387 | 0.407 | 0.414 | 0.427 | 0.386 | 0.402 | 0.374 | 0.399 | 0.437 | 0.449 |
| ETTm1 | 96 | 0.333 | **0.351** | 0.330 | 0.357 | 0.334 | 0.368 | **0.329** | 0.367 | 0.338 | 0.375 | 0.345 | 0.355 | 0.355 | 0.376 | 0.379 | 0.419 |
| | 192 | 0.376 | **0.374** | 0.372 | 0.381 | 0.377 | 0.391 | **0.367** | 0.385 | 0.374 | 0.387 | 0.390 | 0.377 | 0.391 | 0.392 | 0.426 | 0.441 |
| | 336 | 0.404 | **0.395** | 0.408 | 0.405 | 0.426 | 0.420 | **0.399** | 0.410 | 0.410 | 0.411 | 0.425 | 0.399 | 0.424 | 0.415 | 0.445 | 0.459 |
| | 720 | 0.463 | **0.429** | 0.467 | 0.440 | 0.491 | 0.459 | **0.454** | 0.439 | 0.478 | 0.450 | 0.496 | 0.438 | 0.487 | 0.450 | 0.543 | 0.490 |
| | Avg. | 0.399 | **0.390** | 0.394 | 0.396 | 0.407 | 0.410 | **0.387** | 0.400 | 0.400 | 0.406 | 0.414 | 0.392 | 0.414 | 0.408 | 0.448 | 0.452 |
| ETTm2 | 96 | 0.178 | **0.258** | 0.184 | 0.268 | 0.180 | 0.264 | **0.175** | 0.259 | 0.187 | 0.267 | 0.184 | 0.264 | 0.182 | 0.265 | 0.203 | 0.287 |
| | 192 | **0.234** | **0.298** | 0.257 | 0.323 | 0.250 | 0.309 | 0.241 | 0.302 | 0.249 | 0.309 | 0.248 | 0.304 | 0.246 | 0.304 | 0.269 | 0.328 |
| | 336 | **0.294** | **0.333** | 0.324 | 0.367 | 0.311 | 0.348 | 0.305 | 0.343 | 0.321 | 0.351 | 0.308 | 0.338 | 0.307 | 0.342 | 0.325 | 0.366 |
| | 720 | **0.383** | **0.386** | 0.405 | 0.412 | 0.412 | 0.407 | 0.402 | 0.400 | 0.408 | 0.403 | 0.407 | 0.396 | 0.407 | 0.398 | 0.421 | 0.415 |
| | Avg. | **0.272** | **0.319** | 0.293 | 0.343 | 0.288 | 0.332 | 0.281 | 0.326 | 0.291 | 0.333 | 0.287 | 0.326 | 0.286 | 0.327 | 0.305 | 0.349 |
| Electricity | 96 | 0.200 | 0.271 | **0.143** | **0.227** | 0.148 | 0.240 | 0.181 | 0.270 | 0.168 | 0.272 | 0.206 | 0.274 | 0.201 | 0.281 | 0.193 | 0.308 |
| | 192 | 0.198 | 0.274 | **0.155** | **0.245** | 0.162 | 0.253 | 0.188 | 0.274 | 0.184 | 0.289 | 0.203 | 0.276 | 0.201 | 0.283 | 0.201 | 0.315 |
| | 336 | 0.212 | 0.289 | **0.167** | **0.259** | 0.178 | 0.269 | 0.204 | 0.293 | 0.198 | 0.300 | 0.216 | 0.291 | 0.215 | 0.298 | 0.214 | 0.329 |
| | 720 | 0.252 | 0.322 | **0.189** | **0.282** | 0.225 | 0.317 | 0.246 | 0.324 | 0.220 | 0.320 | 0.257 | 0.325 | 0.257 | 0.331 | 0.246 | 0.355 |
| | Avg. | 0.215 | 0.289 | **0.164** | **0.253** | 0.178 | 0.270 | 0.205 | 0.290 | 0.193 | 0.295 | 0.221 | 0.291 | 0.219 | 0.298 | 0.214 | 0.327 |
| Traffic | 96 | 0.662 | 0.366 | 0.429 | 0.251 | **0.395** | **0.268** | 0.462 | 0.290 | 0.593 | 0.321 | 0.705 | 0.372 | 0.649 | 0.389 | 0.587 | 0.366 |
| | 192 | 0.621 | 0.346 | 0.449 | 0.258 | **0.417** | 0.276 | 0.466 | 0.290 | 0.617 | 0.336 | 0.656 | 0.349 | 0.601 | 0.366 | 0.604 | 0.373 |
| | 336 | 0.634 | 0.345 | 0.469 | 0.269 | **0.433** | 0.283 | 0.482 | 0.300 | 0.629 | 0.336 | 0.657 | 0.351 | 0.609 | 0.369 | 0.621 | 0.383 |
| | 720 | 0.654 | 0.364 | 0.503 | 0.287 | **0.467** | 0.302 | 0.514 | 0.320 | 0.640 | 0.350 | 0.688 | 0.372 | 0.647 | 0.387 | 0.626 | 0.382 |
| | Avg. | 0.643 | 0.355 | 0.463 | 0.266 | **0.428** | 0.282 | 0.481 | 0.300 | 0.620 | 0.336 | 0.676 | 0.361 | 0.627 | 0.378 | 0.610 | 0.376 |
| Weather | 96 | 0.199 | 0.229 | **0.160** | **0.207** | 0.174 | 0.214 | 0.177 | 0.210 | 0.172 | 0.220 | 0.224 | 0.254 | 0.192 | 0.232 | 0.217 | 0.296 |
| | 192 | 0.234 | 0.261 | **0.205** | 0.253 | 0.221 | 0.254 | 0.225 | **0.250** | 0.219 | 0.261 | 0.263 | 0.285 | 0.240 | 0.271 | 0.276 | 0.336 |
| | 336 | 0.285 | 0.297 | **0.253** | **0.285** | 0.278 | 0.296 | 0.278 | 0.290 | 0.280 | 0.306 | 0.314 | 0.319 | 0.292 | 0.307 | 0.339 | 0.380 |
| | 720 | 0.352 | 0.344 | **0.323** | **0.338** | 0.358 | 0.349 | 0.354 | 0.340 | 0.365 | 0.359 | 0.386 | 0.363 | 0.364 | 0.353 | 0.403 | 0.428 |
| | Avg. | 0.268 | 0.283 | **0.235** | **0.271** | 0.258 | 0.278 | 0.259 | 0.273 | 0.259 | 0.287 | 0.297 | 0.305 | 0.272 | 0.291 | 0.309 | 0.360 |

Table 12: Multivariate long-term forecasting results of PTN-RLI (i.e. PTN with RLINEAR backbone) and PTN-ITR (i.e. PTN with ITRANSFORMER backbone). We use look-back length fixed as **512** and prediction lengths $T \in \{96, 192, 336, 720\}$ for all datasets. The best results are in **bold** and the second best is underlined.

| Models | | PTN-RLI | | PTN-ITR | | ITRANSFORMER | | PATCHTST | | TIMESNET | | SPARSETSF | | DLINEAR | | FEDFORMER | |
|---|---|---|---|---|---|---|---|---|---|---|---|---|---|---|---|---|---|
| Metric | | MSE | MAE | MSE | MAE | MSE | MAE | MSE | MAE | MSE | MAE | MSE | MAE | MSE | MAE | MSE | MAE |
| ETTh1 | 96 | **0.357** | **0.383** | 0.413 | 0.435 | 0.421 | 0.442 | 0.375 | 0.397 | 0.384 | 0.402 | _0.362_ | _0.388_ | 0.382 | 0.405 | 0.376 | 0.419 |
| | 192 | **0.397** | **0.407** | 0.439 | 0.450 | 0.448 | 0.454 | 0.412 | 0.421 | 0.436 | 0.429 | _0.403_ | _0.411_ | 0.417 | 0.425 | 0.420 | 0.448 |
| | 336 | **0.427** | **0.426** | 0.468 | 0.474 | 0.452 | 0.462 | 0.436 | 0.437 | 0.491 | 0.469 | _0.434_ | _0.428_ | 0.436 | 0.442 | 0.459 | 0.465 |
| | 720 | 0.431 | _0.450_ | 0.542 | 0.546 | 0.564 | 0.558 | 0.465 | 0.471 | 0.521 | 0.500 | **0.426** | **0.447** | 0.433 | 0.455 | 0.506 | 0.507 |
| | Avg. | **0.403** | **0.417** | 0.466 | 0.476 | 0.471 | 0.479 | 0.422 | 0.431 | 0.458 | 0.450 | _0.406_ | _0.419_ | 0.417 | 0.432 | 0.440 | 0.460 |
| ETTh2 | 96 | **0.268** | **0.326** | 0.322 | 0.375 | 0.353 | 0.387 | 0.287 | _0.342_ | 0.340 | 0.374 | 0.294 | 0.346 | _0.272_ | 0.336 | 0.358 | 0.397 |
| | 192 | 0.334 | **0.371** | 0.379 | 0.410 | 0.444 | 0.449 | 0.349 | 0.383 | 0.402 | 0.414 | _0.339_ | _0.377_ | **0.333** | 0.375 | 0.429 | 0.439 |
| | 336 | 0.373 | 0.403 | 0.442 | 0.452 | 0.434 | 0.451 | 0.364 | 0.404 | 0.452 | 0.452 | _0.359_ | _0.397_ | **0.355** | **0.396** | 0.496 | 0.487 |
| | 720 | 0.417 | 0.441 | 0.489 | 0.487 | 0.545 | 0.520 | 0.411 | 0.437 | 0.462 | 0.468 | _0.383_ | _0.424_ | **0.378** | **0.423** | 0.463 | 0.474 |
| | Avg. | 0.348 | 0.385 | 0.408 | 0.431 | 0.444 | 0.452 | 0.353 | 0.392 | 0.414 | 0.427 | _0.344_ | _0.386_ | **0.335** | **0.383** | 0.437 | 0.449 |
| ETTm1 | 96 | **0.293** | **0.336** | 0.322 | 0.354 | 0.324 | 0.363 | _0.297_ | _0.337_ | 0.338 | 0.375 | 0.312 | 0.354 | 0.311 | 0.354 | 0.364 | 0.416 |
| | 192 | **0.330** | **0.358** | 0.371 | 0.394 | 0.377 | 0.396 | _0.340_ | _0.366_ | 0.374 | 0.387 | 0.347 | 0.376 | _0.340_ | 0.369 | 0.426 | 0.441 |
| | 336 | **0.361** | **0.378** | 0.399 | 0.412 | 0.400 | 0.409 | 0.375 | 0.386 | 0.410 | 0.411 | _0.367_ | 0.386 | _0.367_ | _0.385_ | 0.445 | 0.459 |
| | 720 | 0.420 | 0.411 | 0.452 | 0.441 | 0.455 | 0.443 | 0.424 | 0.419 | 0.478 | 0.450 | _0.419_ | 0.413 | **0.416** | _0.412_ | 0.543 | 0.490 |
| | Avg. | **0.351** | **0.371** | 0.386 | 0.400 | 0.389 | 0.403 | 0.380 | 0.390 | 0.400 | 0.406 | 0.361 | 0.382 | _0.359_ | _0.380_ | 0.448 | 0.452 |
| ETTm2 | 96 | **0.162** | **0.248** | 0.189 | 0.275 | 0.186 | 0.272 | 0.176 | 0.259 | 0.187 | 0.267 | _0.163_ | _0.252_ | _0.163_ | 0.254 | 0.203 | 0.287 |
| | 192 | **0.216** | **0.285** | 0.336 | 0.375 | 0.250 | 0.315 | 0.237 | 0.304 | 0.249 | 0.309 | _0.217_ | _0.290_ | _0.217_ | 0.291 | 0.269 | 0.328 |
| | 336 | _0.269_ | **0.321** | 0.320 | 0.360 | 0.303 | 0.347 | 0.301 | 0.346 | 0.321 | 0.351 | 0.270 | 0.327 | **0.268** | _0.326_ | 0.325 | 0.366 |
| | 720 | 0.357 | **0.377** | 0.403 | 0.412 | 0.395 | 0.407 | 0.409 | 0.408 | 0.408 | 0.403 | _0.352_ | 0.379 | **0.349** | _0.378_ | 0.421 | 0.415 |
| | Avg. | _0.251_ | **0.308** | 0.312 | 0.356 | 0.283 | 0.335 | 0.281 | 0.329 | 0.291 | 0.333 | _0.251_ | _0.312_ | **0.249** | _0.312_ | 0.305 | 0.349 |
| Electricity | 96 | 0.135 | _0.230_ | **0.129** | **0.221** | 0.130 | 0.222 | _0.133_ | 0.224 | 0.168 | 0.272 | 0.138 | 0.233 | 0.145 | 0.248 | 0.193 | 0.308 |
| | 192 | _0.149_ | 0.243 | _0.149_ | _0.240_ | 0.149 | 0.242 | **0.148** | **0.237** | 0.184 | 0.289 | 0.151 | 0.244 | 0.159 | 0.260 | 0.201 | 0.315 |
| | 336 | **0.165** | _0.259_ | 0.167 | _0.259_ | 0.170 | 0.263 | 0.165 | **0.256** | 0.198 | 0.300 | 0.166 | 0.260 | 0.175 | 0.275 | 0.214 | 0.329 |
| | 720 | _0.203_ | 0.290 | **0.188** | **0.278** | 0.196 | _0.288_ | _0.203_ | 0.287 | 0.220 | 0.320 | 0.205 | 0.293 | 0.212 | 0.305 | 0.246 | 0.355 |
| | Avg. | 0.163 | 0.255 | **0.158** | **0.250** | _0.161_ | 0.254 | 0.162 | _0.251_ | 0.193 | 0.295 | 0.165 | 0.258 | 0.173 | 0.272 | 0.214 | 0.327 |
| Traffic | 96 | 0.414 | 0.269 | _0.370_ | **0.234** | **0.349** | _0.243_ | 0.397 | 0.259 | 0.593 | 0.321 | 0.389 | 0.268 | 0.398 | 0.286 | 0.587 | 0.366 |
| | 192 | 0.422 | 0.271 | _0.394_ | **0.245** | **0.363** | _0.246_ | 0.410 | 0.266 | 0.617 | 0.336 | 0.400 | 0.272 | 0.409 | 0.289 | 0.604 | 0.373 |
| | 336 | 0.433 | 0.276 | 0.415 | 0.256 | **0.381** | _0.258_ | 0.417 | 0.269 | 0.629 | 0.336 | _0.411_ | 0.275 | 0.421 | 0.294 | 0.621 | 0.383 |
| | 720 | 0.466 | 0.293 | 0.451 | **0.274** | **0.425** | _0.282_ | _0.451_ | 0.287 | 0.640 | 0.350 | _0.448_ | 0.297 | 0.457 | 0.311 | 0.626 | 0.382 |
| | Avg. | 0.434 | 0.277 | _0.408_ | **0.252** | **0.380** | _0.257_ | 0.419 | 0.270 | 0.620 | 0.336 | 0.412 | 0.278 | 0.421 | 0.295 | 0.610 | 0.376 |
| Weather | 96 | 0.162 | 0.209 | 0.163 | _0.208_ | _0.158_ | 0.213 | **0.154** | **0.190** | 0.172 | 0.220 | 0.169 | 0.223 | 0.170 | 0.225 | 0.217 | 0.296 |
| | 192 | _0.205_ | _0.246_ | 0.216 | 0.254 | 0.209 | 0.262 | **0.204** | **0.239** | 0.219 | 0.261 | 0.214 | 0.262 | 0.212 | 0.260 | 0.276 | 0.336 |
| | 336 | **0.250** | **0.282** | 0.271 | 0.296 | 0.253 | 0.295 | 0.268 | 0.286 | 0.280 | 0.306 | _0.257_ | 0.293 | 0.258 | 0.294 | 0.339 | 0.380 |
| | 720 | **0.318** | **0.333** | 0.382 | 0.366 | 0.319 | 0.347 | 0.354 | 0.344 | 0.365 | 0.359 | _0.321_ | 0.340 | 0.320 | 0.339 | 0.403 | 0.428 |
| | Avg. | **0.234** | _0.268_ | 0.258 | 0.281 | _0.235_ | 0.279 | 0.245 | **0.265** | 0.259 | 0.287 | 0.240 | 0.280 | 0.240 | 0.280 | 0.309 | 0.360 |

Table 13: Full results for performance comparisons between backbone models and them integrated with our PTN on ETT, electricity, traffic and weather datasets.

| Models | LINEAR | | +PTN | | DLINEAR | | +PTN | | RLINEAR | | +PTN | | iTRANSFORMER | | +PTN | | PatchTST | | +PTN | |
|---|---|---|---|---|---|---|---|---|---|---|---|---|---|---|---|---|---|---|---|---|
| Metric | MSE | MAE | MSE | MAE | MSE | MAE | MSE | MAE | MSE | MAE | MSE | MAE | MSE | MAE | MSE | MAE | MSE | MAE | MSE | MAE |
| ETTh1 96 | **0.379** | 0.394 | 0.381 | **0.391** | 0.386 | 0.395 | **0.373** | **0.387** | 0.383 | 0.386 | **0.372** | **0.385** | 0.386 | 0.405 | 0.389 | 0.407 | 0.414 | 0.419 | **0.383** | **0.401** |
| ETTh1 192 | **0.430** | 0.425 | 0.444 | 0.434 | 0.437 | 0.424 | **0.431** | **0.419** | 0.437 | **0.417** | **0.431** | 0.418 | 0.441 | 0.436 | **0.432** | **0.432** | 0.460 | 0.445 | **0.429** | **0.424** |
| ETTh1 336 | 0.474 | 0.452 | 0.477 | **0.449** | 0.479 | 0.446 | **0.469** | **0.440** | 0.482 | 0.440 | **0.463** | **0.438** | 0.487 | 0.458 | **0.460** | **0.443** | 0.501 | 0.466 | **0.473** | **0.450** |
| ETTh1 720 | 0.505 | 0.504 | **0.501** | **0.497** | 0.481 | 0.470 | **0.468** | **0.462** | 0.476 | 0.460 | **0.464** | **0.460** | 0.503 | 0.491 | 0.532 | 0.523 | 0.500 | 0.488 | **0.495** | **0.494** |
| ETTh1 Avg. | **0.447** | 0.444 | 0.451 | **0.443** | 0.446 | 0.434 | **0.435** | **0.427** | 0.444 | 0.426 | **0.437** | 0.427 | 0.454 | 0.448 | **0.453** | 0.451 | 0.469 | 0.455 | **0.445** | **0.442** |
| ETTh2 96 | 0.294 | 0.346 | **0.283** | **0.335** | 0.288 | 0.338 | **0.287** | **0.336** | 0.291 | 0.335 | **0.281** | **0.330** | 0.297 | 0.349 | 0.323 | 0.367 | 0.302 | 0.348 | **0.294** | **0.343** |
| ETTh2 192 | 0.380 | 0.401 | **0.370** | **0.394** | 0.374 | 0.390 | **0.367** | **0.385** | 0.376 | 0.387 | **0.361** | **0.380** | 0.380 | 0.400 | **0.377** | **0.399** | 0.388 | 0.400 | **0.379** | **0.400** |
| ETTh2 336 | 0.449 | 0.453 | **0.450** | **0.451** | 0.415 | 0.426 | 0.417 | **0.423** | 0.420 | 0.423 | **0.409** | **0.418** | 0.428 | 0.432 | 0.453 | 0.450 | **0.426** | **0.433** | 0.450 | 0.443 |
| ETTh2 720 | 0.588 | 0.541 | **0.580** | **0.534** | 0.420 | 0.440 | 0.436 | 0.454 | 0.418 | 0.435 | **0.413** | **0.434** | 0.427 | 0.445 | 0.610 | 0.552 | 0.431 | 0.446 | 0.474 | 0.472 |
| ETTh2 Avg. | 0.428 | 0.435 | **0.421** | **0.429** | 0.374 | 0.399 | 0.377 | 0.400 | 0.376 | 0.395 | **0.366** | **0.391** | 0.383 | 0.407 | 0.441 | 0.442 | **0.387** | **0.407** | 0.399 | 0.415 |
| ETTm1 96 | 0.338 | 0.360 | **0.328** | **0.349** | 0.355 | 0.376 | **0.335** | **0.354** | 0.337 | 0.352 | **0.333** | **0.351** | 0.334 | 0.368 | **0.330** | **0.357** | 0.329 | 0.367 | **0.315** | **0.344** |
| ETTm1 192 | 0.379 | 0.382 | **0.374** | **0.374** | 0.391 | 0.392 | **0.379** | **0.376** | 0.383 | 0.375 | **0.376** | **0.374** | 0.377 | 0.391 | **0.372** | **0.381** | 0.367 | 0.385 | **0.363** | **0.373** |
| ETTm1 336 | 0.410 | 0.406 | **0.405** | **0.396** | 0.424 | 0.415 | **0.413** | **0.398** | 0.417 | 0.397 | **0.404** | **0.395** | 0.426 | 0.420 | **0.408** | **0.405** | 0.399 | 0.410 | **0.389** | **0.392** |
| ETTm1 720 | 0.472 | 0.445 | **0.470** | **0.437** | 0.487 | 0.450 | **0.475** | **0.434** | 0.485 | 0.435 | **0.463** | **0.429** | 0.491 | 0.459 | **0.467** | **0.440** | 0.454 | 0.439 | **0.457** | **0.434** |
| ETTm1 Avg. | 0.400 | 0.398 | **0.394** | **0.389** | 0.414 | 0.408 | **0.401** | **0.390** | 0.405 | 0.390 | **0.399** | **0.390** | 0.407 | 0.410 | **0.394** | **0.396** | 0.387 | 0.400 | **0.381** | **0.386** |
| ETTm2 96 | 0.183 | 0.263 | **0.179** | **0.260** | 0.182 | 0.265 | **0.179** | **0.258** | 0.183 | 0.257 | **0.178** | **0.258** | 0.180 | 0.264 | 0.184 | 0.268 | 0.175 | 0.259 | **0.171** | **0.258** |
| ETTm2 192 | 0.245 | 0.307 | **0.238** | **0.300** | 0.246 | 0.304 | **0.239** | **0.302** | 0.247 | 0.299 | **0.234** | **0.298** | 0.250 | 0.309 | 0.257 | 0.323 | 0.241 | 0.302 | **0.231** | **0.300** |
| ETTm2 336 | 0.306 | 0.349 | **0.297** | **0.342** | 0.307 | **0.342** | **0.296** | 0.336 | 0.307 | 0.337 | **0.294** | **0.333** | 0.311 | 0.348 | 0.324 | 0.367 | **0.305** | **0.343** | 0.344 | 0.343 |
| ETTm2 720 | 0.403 | 0.407 | **0.392** | **0.397** | 0.407 | 0.398 | **0.391** | **0.396** | 0.408 | 0.394 | **0.383** | **0.386** | 0.412 | 0.407 | 0.405 | 0.412 | **0.402** | **0.400** | 0.424 | 0.421 |
| ETTm2 Avg. | 0.284 | 0.332 | **0.277** | **0.325** | 0.286 | 0.327 | **0.276** | **0.323** | 0.286 | 0.322 | **0.272** | **0.319** | 0.288 | 0.332 | 0.293 | 0.343 | **0.281** | **0.326** | 0.293 | 0.331 |
| Electricity 96 | 0.200 | 0.277 | **0.197** | **0.271** | 0.201 | 0.281 | **0.199** | **0.271** | 0.201 | 0.270 | **0.200** | 0.271 | 0.148 | 0.240 | **0.143** | **0.227** | 0.181 | 0.270 | **0.169** | **0.248** |
| Electricity 192 | 0.199 | 0.281 | **0.196** | **0.274** | 0.201 | 0.283 | **0.198** | **0.274** | 0.200 | 0.273 | 0.200 | 0.274 | 0.162 | 0.253 | **0.155** | **0.245** | 0.188 | 0.274 | **0.177** | **0.257** |
| Electricity 336 | 0.212 | 0.297 | **0.209** | **0.289** | 0.215 | 0.298 | **0.211** | **0.289** | 0.214 | 0.289 | **0.212** | **0.289** | 0.178 | 0.269 | **0.167** | **0.259** | 0.204 | 0.293 | **0.194** | **0.275** |
| Electricity 720 | 0.248 | 0.328 | **0.244** | **0.320** | 0.257 | 0.331 | **0.251** | **0.321** | 0.255 | 0.322 | **0.252** | **0.322** | 0.225 | 0.317 | **0.189** | **0.282** | 0.246 | 0.324 | **0.242** | **0.318** |
| Electricity Avg. | 0.215 | 0.296 | **0.212** | **0.289** | 0.219 | 0.298 | **0.215** | **0.289** | 0.217 | 0.289 | **0.215** | **0.289** | 0.178 | 0.270 | **0.164** | **0.253** | 0.205 | 0.290 | **0.196** | **0.274** |
| Traffic 96 | **0.653** | 0.376 | 0.702 | **0.366** | **0.649** | 0.389 | 0.670 | **0.365** | **0.662** | 0.366 | 0.694 | **0.365** | **0.395** | 0.268 | 0.429 | **0.251** | **0.462** | 0.290 | 0.548 | **0.281** |
| Traffic 192 | **0.603** | 0.351 | 0.657 | **0.343** | **0.601** | 0.366 | 0.625 | **0.343** | **0.621** | 0.346 | 0.651 | **0.342** | **0.417** | 0.276 | 0.449 | **0.258** | **0.466** | 0.290 | 0.554 | **0.277** |
| Traffic 336 | **0.607** | 0.353 | 0.666 | **0.348** | **0.609** | 0.369 | 0.634 | **0.345** | **0.634** | 0.345 | 0.655 | **0.344** | **0.433** | 0.283 | 0.469 | **0.269** | **0.482** | 0.300 | 0.576 | **0.288** |
| Traffic 720 | **0.642** | 0.373 | 0.674 | **0.352** | **0.647** | 0.387 | 0.667 | **0.364** | **0.654** | 0.364 | 0.686 | **0.364** | **0.467** | 0.302 | 0.503 | **0.287** | **0.514** | 0.320 | 0.659 | **0.330** |
| Traffic Avg. | **0.626** | 0.363 | 0.675 | **0.352** | **0.627** | 0.378 | 0.649 | **0.354** | **0.643** | 0.355 | 0.672 | **0.354** | **0.428** | 0.282 | 0.463 | **0.266** | **0.481** | 0.300 | 0.584 | **0.294** |
| Weather 96 | 0.200 | 0.237 | **0.192** | **0.228** | 0.192 | 0.232 | **0.194** | **0.227** | 0.209 | 0.230 | **0.199** | **0.229** | 0.174 | 0.214 | **0.160** | **0.207** | 0.177 | 0.210 | **0.168** | **0.208** |
| Weather 192 | 0.238 | 0.275 | **0.238** | **0.274** | 0.240 | 0.271 | **0.234** | **0.261** | 0.253 | 0.266 | **0.234** | **0.261** | 0.221 | 0.254 | **0.205** | **0.253** | 0.225 | 0.250 | **0.214** | **0.252** |
| Weather 336 | 0.281 | 0.310 | **0.283** | **0.315** | 0.292 | 0.307 | **0.278** | **0.296** | 0.302 | 0.302 | **0.285** | **0.297** | 0.278 | 0.296 | **0.253** | **0.285** | 0.278 | 0.290 | **0.261** | **0.289** |
| Weather 720 | 0.341 | 0.357 | **0.339** | **0.353** | 0.364 | 0.353 | **0.352** | **0.344** | 0.374 | 0.350 | **0.352** | **0.344** | 0.358 | 0.349 | **0.323** | **0.338** | 0.354 | 0.340 | **0.343** | **0.341** |
| Weather Avg. | 0.265 | 0.295 | **0.263** | **0.292** | 0.272 | 0.291 | **0.265** | **0.282** | 0.284 | 0.287 | **0.268** | **0.283** | 0.258 | 0.278 | **0.235** | **0.271** | 0.259 | 0.273 | **0.246** | **0.272** |

Table 14: Full results for performance comparisons between backbone models and them integrated with our PTN on PeMS datasets.

| Models | DLINEAR | | +PTN | | PatchTST | | +PTN | | iTRANSFORMER | | +PTN | |
|---|---|---|---|---|---|---|---|---|---|---|---|---|
| Metric | MSE | MAE | MSE | MAE | MSE | MAE | MSE | MAE | MSE | MAE | MSE | MAE |
| PEMS03 12 | 0.116 | 0.224 | **0.097** | **0.207** | 0.113 | 0.229 | **0.076** | **0.179** | 0.071 | 0.174 | **0.069** | **0.171** |
| PEMS03 24 | 0.233 | 0.316 | **0.143** | **0.254** | 0.211 | 0.313 | **0.110** | **0.218** | 0.093 | 0.201 | **0.091** | **0.195** |
| PEMS03 36 | 0.381 | 0.410 | **0.202** | **0.304** | 0.362 | 0.420 | **0.143** | **0.251** | 0.125 | 0.236 | **0.115** | **0.222** |
| PEMS03 48 | 0.536 | 0.505 | **0.246** | **0.345** | 0.504 | 0.512 | **0.164** | **0.271** | 0.164 | 0.275 | **0.135** | **0.240** |
| PEMS03 Avg. | 0.317 | 0.364 | **0.172** | **0.277** | 0.298 | 0.368 | **0.123** | **0.230** | 0.113 | 0.222 | **0.102** | **0.207** |
| PEMS04 12 | 0.128 | 0.236 | **0.107** | **0.215** | 0.128 | 0.244 | **0.094** | **0.198** | 0.078 | 0.183 | 0.082 | **0.183** |
| PEMS04 24 | 0.243 | 0.329 | **0.161** | **0.269** | 0.252 | 0.355 | **0.143** | **0.248** | 0.095 | 0.205 | 0.107 | 0.213 |
| PEMS04 36 | 0.390 | 0.424 | **0.238** | **0.330** | 0.413 | 0.472 | **0.172** | **0.276** | 0.120 | 0.233 | 0.134 | 0.240 |
| PEMS04 48 | 0.555 | 0.518 | **0.266** | **0.352** | 0.573 | 0.573 | **0.202** | **0.305** | 0.150 | 0.262 | 0.161 | 0.266 |
| PEMS04 Avg. | 0.329 | 0.377 | **0.193** | **0.292** | 0.341 | 0.411 | **0.153** | **0.257** | 0.111 | 0.221 | 0.121 | 0.225 |
| PEMS07 12 | 0.109 | 0.219 | **0.096** | **0.203** | 0.099 | 0.212 | **0.075** | **0.167** | 0.067 | 0.165 | **0.066** | **0.158** |
| PEMS07 24 | 0.230 | 0.318 | **0.160** | **0.265** | 0.203 | 0.311 | **0.115** | **0.214** | 0.088 | 0.190 | 0.089 | **0.184** |
| PEMS07 36 | 0.385 | 0.419 | **0.209** | **0.308** | 0.333 | 0.406 | **0.135** | **0.236** | 0.110 | 0.215 | 0.113 | **0.209** |
| PEMS07 48 | 0.551 | 0.515 | **0.262** | **0.355** | 0.535 | 0.536 | **0.156** | **0.255** | 0.139 | 0.245 | **0.134** | **0.226** |
| PEMS07 Avg. | 0.319 | 0.368 | **0.182** | **0.283** | 0.293 | 0.366 | **0.120** | **0.218** | 0.101 | 0.204 | **0.101** | **0.194** |
| PEMS08 12 | **0.121** | 0.229 | 0.152 | 0.223 | **0.120** | 0.239 | 0.140 | 0.198 | 0.079 | 0.182 | 0.128 | 0.184 |
| PEMS08 24 | **0.235** | 0.321 | 0.242 | 0.292 | 0.226 | 0.331 | **0.187** | 0.240 | 0.115 | 0.219 | 0.161 | 0.212 |
| PEMS08 36 | **0.380** | 0.419 | 0.318 | **0.351** | 0.345 | 0.414 | **0.230** | **0.276** | 0.186 | 0.235 | 0.188 | 0.233 |
| PEMS08 48 | 0.549 | 0.518 | **0.325** | **0.368** | 0.506 | 0.502 | **0.245** | **0.294** | 0.221 | 0.267 | 0.215 | 0.252 |
| PEMS08 Avg. | 0.321 | 0.372 | **0.259** | **0.308** | 0.299 | 0.372 | **0.200** | **0.252** | 0.150 | 0.226 | 0.173 | 0.220 |

## A.7 MORE VISUALIZATIONS

### A.7.1 WEIGHT SPARSITY COMPARISONS

As proposed in SparseTSF (Lin et al., 2024), TSF datasets can require far fewer parameters involved and information provided to make near-SOTA forecasting. The explanation for this might be the sparsity. As some work (Ilbert et al., 2024; Tan et al., 2024) also suggests, complex modeling of Transformer models might suffer from overfitting non-sparse features in time series data that even have degraded Indentity matrix as attention scores. The visualization of weight matrices learned can

Table 15: Full results for Table 4 on comparison of generalization capabilities of our data transfer protocol. 'A→B' indicates the raw data transformation is learned by Model A, and then fed to the Model B

| Models | | PATCHTST | | RLINEAR→PATCHTST | | iTRANSFORMER | | RLINEAR→iTRANSFORMER | |
|---|---|---|---|---|---|---|---|---|---|
| Metric | | MSE | MAE | MSE | MAE | MSE | MAE | MSE | MAE |
| ETTh1 | 96 | 0.414 | 0.419 | **0.402** | **0.410** | 0.386 | **0.405** | **0.395** | 0.409 |
| | 192 | 0.460 | 0.445 | **0.425** | **0.427** | 0.441 | 0.436 | **0.432** | **0.430** |
| | 336 | 0.501 | 0.466 | **0.473** | **0.456** | 0.487 | 0.458 | **0.473** | **0.453** |
| | 720 | 0.500 | 0.488 | **0.515** | **0.505** | 0.503 | **0.491** | 0.524 | 0.513 |
| | Avg. | 0.469 | 0.455 | **0.454** | **0.450** | 0.454 | **0.448** | 0.456 | 0.451 |
| ETTh2 | 96 | 0.302 | 0.348 | **0.306** | **0.349** | **0.297** | **0.349** | 0.372 | 0.400 |
| | 192 | **0.388** | **0.400** | 0.389 | 0.402 | **0.380** | **0.400** | 0.390 | 0.403 |
| | 336 | **0.426** | **0.433** | 0.460 | 0.443 | **0.428** | **0.432** | 0.452 | 0.450 |
| | 720 | **0.431** | **0.446** | 0.527 | 0.526 | **0.427** | **0.445** | 0.777 | 0.619 |
| | Avg. | **0.387** | **0.407** | 0.420 | 0.430 | **0.383** | **0.407** | 0.498 | 0.468 |
| Electricity | 96 | 0.181 | 0.270 | **0.169** | **0.247** | 0.148 | 0.240 | **0.147** | **0.233** |
| | 192 | 0.188 | 0.274 | **0.177** | **0.256** | 0.162 | 0.253 | **0.162** | **0.248** |
| | 336 | 0.204 | 0.293 | **0.192** | **0.272** | 0.178 | 0.269 | **0.172** | **0.262** |
| | 720 | 0.246 | 0.324 | **0.230** | **0.304** | 0.225 | 0.317 | **0.185** | **0.277** |
| | Avg. | 0.205 | 0.290 | **0.192** | **0.270** | 0.178 | 0.270 | **0.166** | **0.255** |
| Traffic | 96 | **0.462** | 0.290 | 0.549 | **0.274** | **0.395** | 0.268 | 0.486 | **0.242** |
| | 192 | **0.466** | 0.290 | 0.552 | **0.277** | **0.417** | 0.276 | 0.510 | **0.253** |
| | 336 | **0.482** | 0.300 | 0.568 | **0.283** | **0.433** | 0.283 | 0.527 | **0.264** |
| | 720 | **0.514** | 0.320 | 0.602 | **0.302** | **0.467** | 0.302 | 0.558 | **0.281** |
| | Avg. | **0.481** | 0.300 | 0.568 | **0.284** | **0.428** | 0.282 | 0.520 | **0.260** |

provide some insights, such as comparisons in sparsity. Here we also show the visualized weight matrices between w. and w./o. PTN in Figures 17-20 as an intuitive comparison.

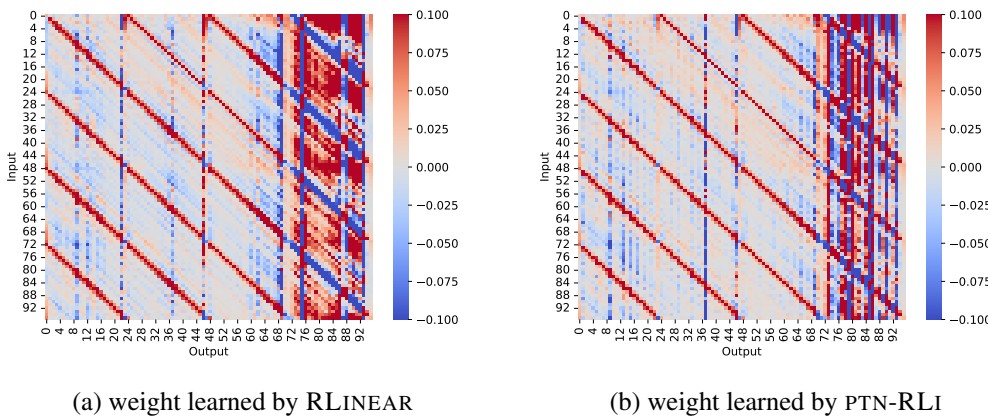

(a) weight learned by RLINEAR                    (b) weight learned by PTN-RLI

Figure 17: Comparision between weights learned on ETTh1.

### A.7.2 TRANING OF THE PROXIMAL TRANSFORMATION

As Section 5.2 showcases, the training processes of how the transformed data approaches the raw data are Channel-dependent. Here, we show all channels in the following Figures 21-27.

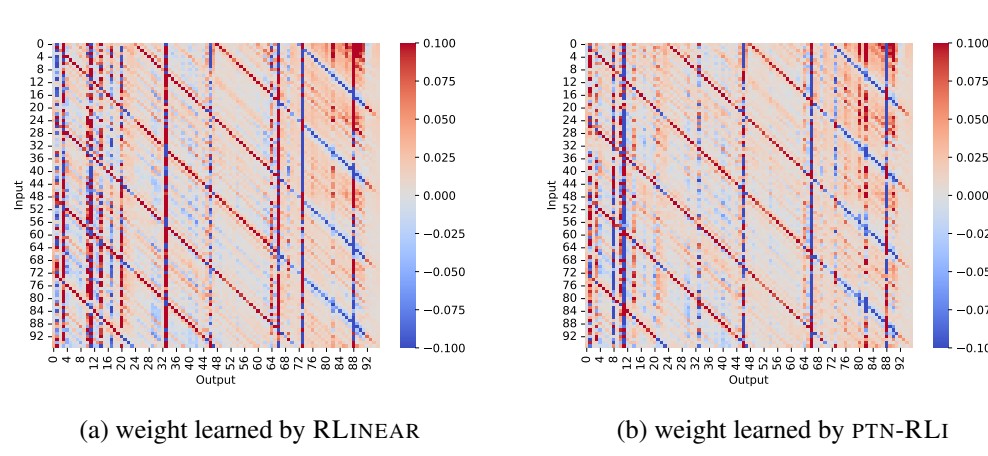

(a) weight learned by RLINEAR                    (b) weight learned by PTN-RLI

Figure 18: Comparision between weights learned on ETTh2.

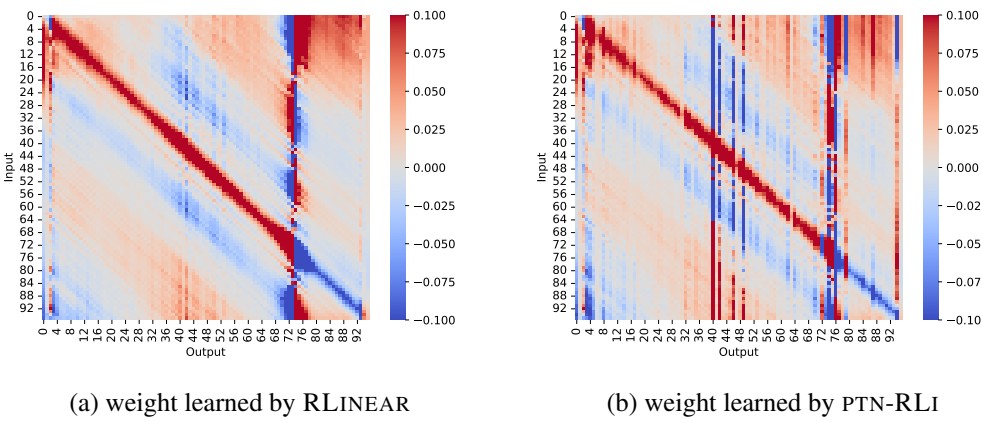

(a) weight learned by RLINEAR                    (b) weight learned by PTN-RLI

Figure 19: Comparision between weights learned on ETTm1.

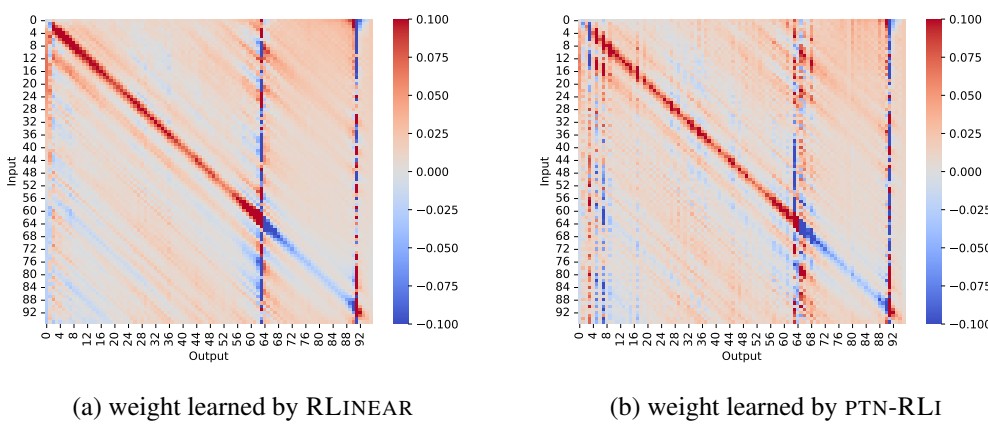

(a) weight learned by RLINEAR                    (b) weight learned by PTN-RLI

Figure 20: Comparision between weights learned on ETTm2.

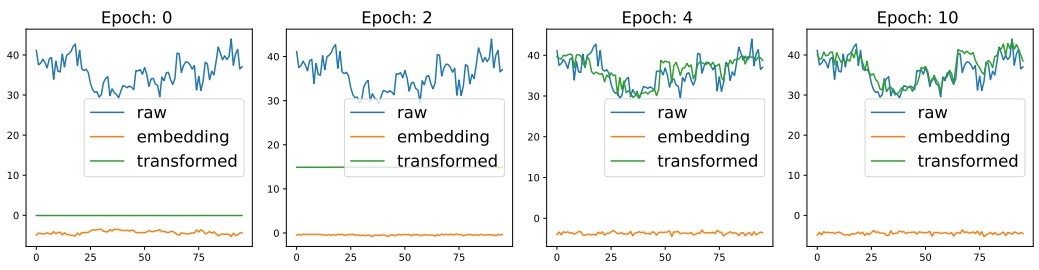

Figure 21: Channel 1 for PTN-RLI on the ETTh2 dataset.

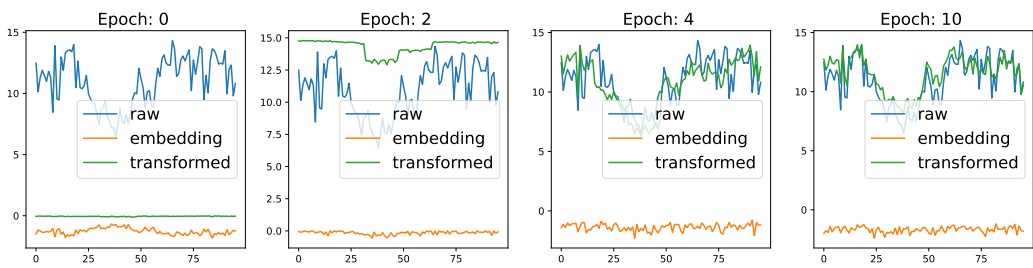

Figure 22: Channel 2 for PTN-RLI on the ETTh2 dataset.

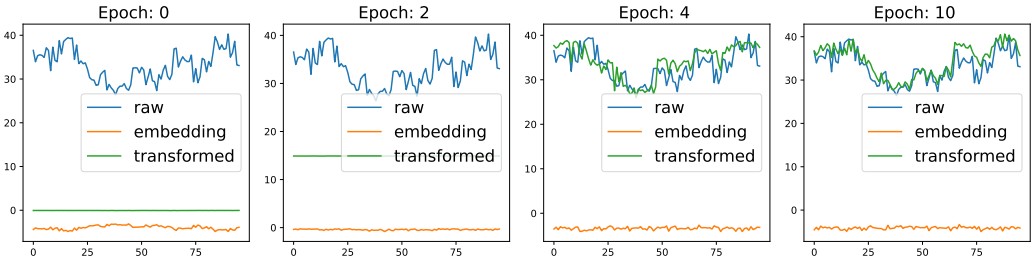

Figure 23: Channel 3 for PTN-RLI on the ETTh2 dataset.

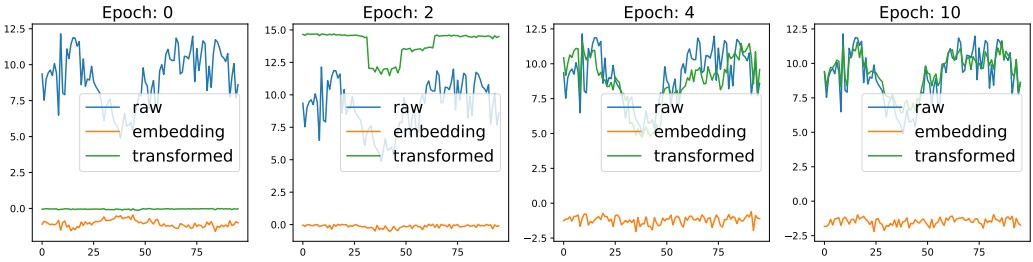

Figure 24: Channel 4 for PTN-RLI on the ETTh2 dataset.

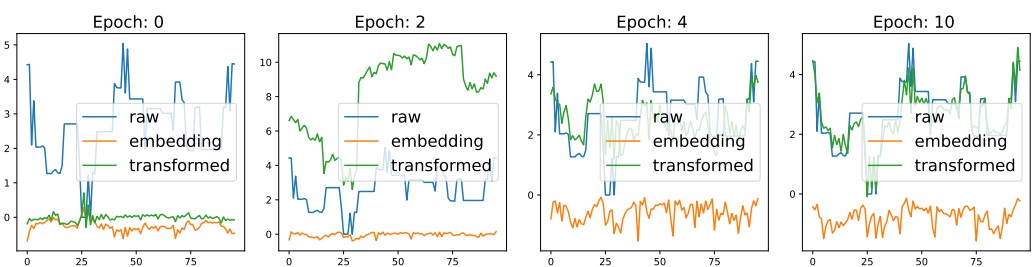

Figure 25: Channel 5 for PTN-RLI on the ETTh2 dataset.

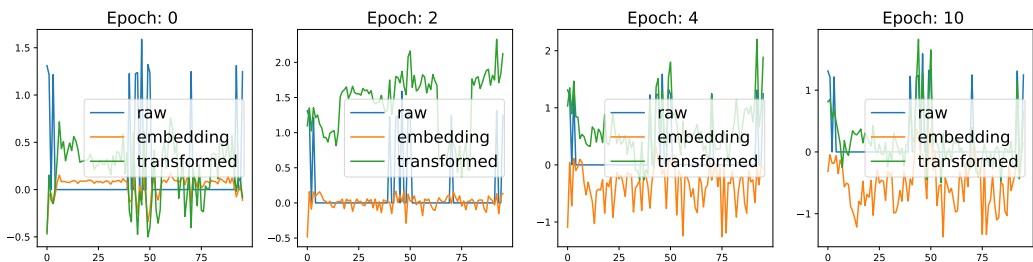

Figure 26: Channel 6 for PTN-RLI on the ETTh2 dataset.

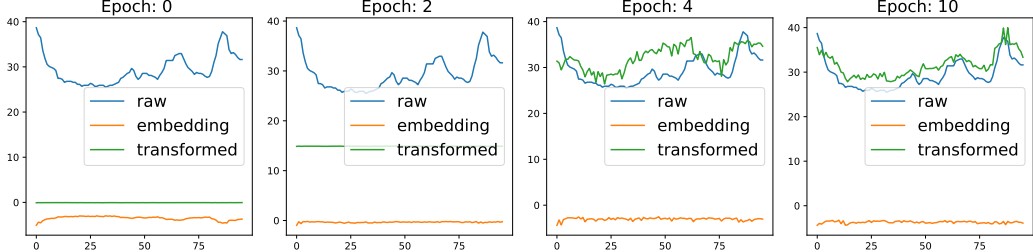

Figure 27: Channel 7 for PTN-RLI on the ETTh2 dataset.

