# OpenReview forum: "Learned Data Transformation: A Data-centric Plugin for Enhancing Time Series Forecasting"
_ICLR.cc/2025/Conference — Submitted to ICLR 2025_

### Official Review · Reviewer_gMsX · 2024-11-03

**Soundness:** 2
**Presentation:** 2
**Contribution:** 2
**Rating:** 5
**Confidence:** 4

**Summary:**

This paper proposes a data transformation model to support long-term time series forecasting. Specifically, it first obtains a transformation of the raw data using Proximal Transformation Networks (PTNs) and then uses the transformed data to train a predictor. Each PTN consists of a convolutional encoder and a decoder with intra-patch attention, channel-wise attention, and a point-wise linear head. Experiments on several benchmark datasets are conducted to evaluate the effectiveness of the proposed model.

**Strengths:**

1. The idea is relatively novel and is supported by some theoretical insights.
2. Extensive experiments from various perspectives are provided.
3. The motivations behind the proposal are well introduced.

**Weaknesses:**

1. The comparison with baselines in Table 9 for look-back length 512 appears to be unfair. For instance, iTransformer should also use a look-back length of 512, and the results of PatchTST in this table are much worse than those reported in the PatchTST paper (PatchTST/64 in Table 3 of its paper).

2. The proposed model does not seem to perform well on complex datasets, such as Traffic. It would be beneficial to provide results on more complex datasets, such as the PEMS datasets used in the iTransformer paper.

3. It would be helpful to include Mean Squared Error (MSE) results in Table 4.

4. It seems inappropriate to claim that MoE is used without a gating network. Additionally, the method for selecting an appropriate number of PTNs, as well as the specific values used in the paper, is unclear.

5. There is no complexity analysis when adding the proposed model to the base models.

6. There are also many unclear points and typos in the paper, such as:

+
Figure 1 is not well explained, e.g., the meaning of "7/8".

+
It is unclear how the outputs of intra-patch attention and channel-wise attention are combined.

+
It is unclear how the prediction process is conducted after training. Should the raw data be directly input to the trained predictor?

+
"An attention-based Encoder" should be "An attention-based Decoder" on Page 2.

+
"Piece-wise Linear Head" should be "Point-wise Linear Head" in Figure 3.

**Questions:**

Same as the weaknesses

---

> ### Author Response · Authors · 2024-11-21
> **Response to W1 and W2**
>
> > [W1-The comparison with baselines in Table 9 for look-back length 512 appears to be unfair. For instance, iTransformer should also use a look-back length of 512, and the results of PatchTST in this table are much worse than those reported in the PatchTST paper (PatchTST/64 in Table 3 of its paper).]
>
> Thank you for raising this concern. We **have updated the iTransformer results with a look-back length of 512 in Table 12 (previously Table 9)**. Notably, we observed **a significant performance drop** on the ETT datasets, likely due to overfitting. This could explain why the original iTransformer paper focuses on using a smaller look-back length of 96, as, perhaps, the architecture was not designed for it.
>
> For clarity, here is a brief overview of the updates made in Table 12:
>
> | **iTransformer, 512 $\to$ {96, 192, 336, 720}** | **ETTh1 (avg)** | **ETTh2 (avg)** | **ETTm1 (avg)** | **ETTm2 (avg)** | **Electricity (avg)** | **Traffic (avg)** | **Weather (avg)** |
> | ----------------------------------------------- | --------------- | --------------- | --------------- | --------------- | --------------------- | ----------------- | ----------------- |
> | MSE                                             | 0.471           | 0.444           | 0.389           | 0.283           | 0.161                 | 0.380             | 0.235             |
> | MAE                                             | 0.479           | 0.452           | 0.403           | 0.335           | 0.254                 | 0.257             | 0.279             |
>
> **The results of PatchTST were impacted by a bug** reported in [1, 2, 3], which is thoroughly discussed at https://github.com/VEWOXIC/FITS. After addressing and fixing this issue, we can confirm that our reported results are credible and reliable.
>
> [1] FITS: Modeling Time Series with 10k Parameters
>
> [2] TFB: Towards Comprehensive and Fair Benchmarking of Time Series Forecasting Methods
>
> [3] SparseTSF: Modeling Long-term Time Series Forecasting with 1k Parameters
>
> > [W2-The proposed model does not seem to perform well on complex datasets, such as Traffic. It would be beneficial to provide results on more complex datasets, such as the PEMS datasets used in the iTransformer paper.]
>
> We have now included the full results on the PeMS datasets in **Table 14**, located in **Appendix A.6**. For clarity, the output lengths differ from those reported in the iTransformer paper. Specifically, we use {12, 24, 36, 48} for our experiments, whereas the original paper lists {12, 24, 48, 96}. This discrepancy arises due to an error in their reporting, as detailed in https://github.com/thuml/iTransformer/issues/91 along with other discussions on reproducing the PeMS results (https://github.com/thuml/iTransformer/issues?q=is%3Aissue+pems). To ensure fair and accurate comparisons, we reran experiments for other backbone models to validate the results.
>
> As briefly shown in table **T3 in "Comments to All"**, our method shows notable improvement on PeMS datasets. iTransformer does not have consistent improvement compared to the other two backbones, which we attribute to its overfitting problem, as also explained in [6].  Additionally, as explained in Section 4.1 in the paragraph starting with **"Training loss"**, our method may struggle in scenarios where the backbone's convergence is not guaranteed. This limitation helps explain iTransformer's occasional degraded performance, such as on PeMS04 and certain ETT datasets, as observed in the main experiments (now in Table 11 and Table 13).

---

> ### Author Response · Authors · 2024-11-21
> **Response to W3 and W4**
>
> > [W3-It would be helpful to include Mean Squared Error (MSE) results in Table 4.]
>
> We have included the full data transfer results, including MSE, in Table 14 in Appendix A.6. Across various datasets (e.g., Traffic, ETTh1/ETTh2 when using longer inputs), MSE is less stable than MAE and can even degrade in some cases, such as Traffic. **This may stem from MSE's sensitivity to smaller values, particularly after standard scaling by mean and variance**. We hope this addition clarifies the challenges of using MSE in these scenarios.
>
> > [W4-It seems inappropriate to claim that MoE is used without a gating network. Additionally, the method for selecting an appropriate number of PTNs, as well as the specific values used in the paper, is unclear.]
>
> In Section 4.2, we described a design that decomposes time series into up to four sub-series, processed by individual "expert" models. These models transform their respective sub-series, and the outputs are concatenated to restore the original length, producing the transformed results.
>
> Our approach shares two similarities with traditional MoE frameworks:
>
> 1. Traditional MoE uses hard-gating or soft-gating [4], with soft-gating assigning weights to expert outputs. Our concatenation of outputs is loosely analogous to a uniform weighting strategy in soft-gating.
> 2. MoE typically balances input distribution across experts, often via auxiliary loss [5]. We achieve a similar balance by manually controlling routing.
>
> We acknowledge that the term "MoE" might cause confusion. A more accurate description would be "parallel version of PTN", reflecting its role in scaling PTN parameters. We will update the terminology in the final version of the paper.
>
> Regarding the selection of the number of experts, we briefly addressed this in Appendix A.5, where we also discussed how MoE applied to data transformation could resemble a decomposition process. To clarify, the MoE-based variation was not employed in the main experiments but rather introduced as an optional acceleration strategy (not listed in our contributions). While not a primary focus, it offers an interesting direction for future research.
>
> [4] From Sparse to Soft Mixtures of Experts
>
> [5] Outrageously Large Neural Networks: The Sparsely-Gated Mixture-of-Experts Layer

---

> ### Author Response · Authors · 2024-11-21
> **Response to W5 and W6**
>
> > [W5-There is no complexity analysis when adding the proposed model to the base models. ]
>
> We have added a "Complexity Analysis" section in **Appendix A.4.3** to discuss the complexity of PTN. By analyzing the core components of PTN, intra-patch and channel-wise attentions, we identified batch size as a previously overlooked factor in complexity analysis (see P1 in Appendix A.4.3). This led us to present more efficient implementations (see P2 and P3 in Appendix A.4.3) that limit parallel computations within a batch. Experimental results (Table 8, Table 9, Figure 10, Figure 11 in the revised paper) demonstrate these implementations reduce PTN's complexity to **match that of linear models**, enhancing its practicality.
>
> These efficient implementations and the corresponding empirical studies are summarized as follows:
>
> - **Mask Fusion** (P2 in Appendix A.4.3): We introduced a method to accelerate attention computations by merging intra-patch and channel-wise attention masks. This reduced complexity from $\mathcal{O}((l_p + C)LC)$ to $\mathcal{O}(l_p^2C^2)$, where $l_p$ (patch length) is as small as 4 in our experiments. Details and results supporting this method are in Appendix A.4.3, Figure 8, and Figure 9, showing its effectiveness in improving PTN's efficiency. A brief view of the results is shown in table **T1 in "Comments to All"** and in Figure 9 in Appendix 4.3.
> - **Variate Sampling** (P3 in Appendix A.4.3): According to our new implementation, adopting the variate sampling technique proposed in iTransformer [1] can also lead to improvement in terms of both training and inference efficiency. The relative increase in time and memory costs can be reduced from several times to a factor of **less than one**. The results are reported in table **T2 in "Comments to All"** and in Tables 8 and 9 in Appendix A.4.3.
>
> In addition to performance-neutral optimizations introduced above, there are also **lossy acceleration methods**. As discussed in Section 5.3, a data transfer method can improve training efficiency by avoiding the direct training of a complex backbone. As inspired by your question if raw is used in inference, although the answer is negative, we reconsider the possibility of using raw data directly as inputs. For inference, we, therefore, propose a method based on train-time distillation that generates a student model without PTN, incurring no additional inference overhead but requiring only the training of an extra backbone. Details are provided in Appendix A.4.4, along with experimental results shown below. However, this approach is not universally applicable and may fail in certain cases, such as the iTransformer on ETT datasets, for reasons outlined in response to W2. The brief results are shown in **T3 in "Comments to All"** and full results are shown in Table 10 in Appendix 4.4.
>
> [1] iTransformer: Inverted Transformers Are Effective for Time Series Forecasting
>
> > [W6-There are also many unclear points and typos in the paper, such as:...]
>
> Thank you for your feedback. Here are our clarifications and updates:
>
> - Figure 1: The "7/8 part" refers to the weights within a window when computing normalization. When calculating $X-\bar{X}$ in a given window, the $i$-th normalized value is derived as $(-\frac{1}{n}, -\frac{1}{n}, \ldots, \frac{n -1}{n}, \ldots, -\frac{1}{n}) * (x_1, x_2, \ldots, x_i, \ldots, x_n)$ (* denotes element-wise product). This approach is unified in a Moving Kernel form with other methods. We have revised Figure 1 and its caption for improved clarity.
> - Attention Outputs: The outputs of the two attentions are added together. Additionally, we have added Figure 8 in Appendix A.4.3 to present a different implementation discussed earlier.
> - Input Data: For the standard version, we use transformed data as input. Inspired by your advice, we are also exploring the possibility of directly using raw data as input, which would incur no additional cost during inference (see Appendix A.4.4 "Efficient Student Model By Train-Time Distillation").
>
> Finally, all other typos and errors have been corrected. Thank you for helping us improve the paper.

---

> > ### Comment · Reviewer_gMsX · 2024-11-25
> >
> > Thank you very much for your responses; I have updated my score accordingly. However, I still have concerns regarding the experiments. For instance, as noted on https://github.com/VEWOXIC/FITS, the discovered bug predominantly affects results on smaller datasets like ETTh1 and ETTh2. It remains unconvincing to observe the significant differences on some other datasets, particularly the Weather dataset, as reflected by the discrepancies between PatchTST in Table 12 of this paper and PatchTST/64 in Table 3 of its original paper. Additionally, Figure 1 is still not well explained and is difficult for readers without prior knowledge to understand.

---

> > > ### Author Response · Authors · 2024-11-25
> > >
> > > Thank you very much for your feedback. We appreciate your focus on the baseline performance. For one thing, we provide codes of what we run for baselines in the anonymous github link at https://anonymous.4open.science/r/PTN-2FC6/ and we welcome to examine the reported performance for PatchTST. For another, we suspect that this is due to the difference of how we treat hyperparameter search from previous works.
> > >
> > > Hyperparameters are very important for TSF, and probably other time series tasks because of the diversity in datasets. However, conducting hyperparameter search on each dataset (probably in most previous works) is very time-consuming, inapplicable in real scenarios.  Therefore, we opt to search hyperparameters on smaller datasets like ETTh1 and ETTh2 and fixed hyperparameters for the rest. This practice is applied to all baselines and our methods to ensure fairness. Certainly, searching small subsets of each dataset would be a more reasonable approach, and we will consider incorporating this practice in, if possible, future version.
> > >
> > > Another point we would like to clarify is, time series datasets are naturally prone to incur different performances. For example, in the PeMS experiments that we recently added, PatchTST has clearly worse performance compared to what is reported in the iTransformer paper. We also have clues on https://github.com/thuml/iTransformer/issues/64 where avoiding using RevIN is suggested to produce superior performance. Since there is no intuition of when to use RevIN on which datasets (though it proves effective for 7 datasets in the main experiments), we consistently apply RevIN for all baselines and our methods to ensure fairness.

---

> > > > ### Comment · Reviewer_gMsX · 2024-11-26
> > > >
> > > > Thank you very much for the further explanation. I would like to keep my updated score based on the current experimental results and presentations.

---

### Official Review · Reviewer_t7bj · 2024-11-03

**Soundness:** 3
**Presentation:** 2
**Contribution:** 2
**Rating:** 5
**Confidence:** 3

**Summary:**

The paper proposes the Proximal Transformation Network (PTN), a plugin for improving time series forecasting (TSF) by learning general, data-centric transformations that enhance model performance while preserving proximity to the original data. PTN, which combines a convolutional encoder and attention-based decoder, can integrate with any TSF model, optimizing both data fidelity and forecasting accuracy. Through experiments on seven real-world datasets, PTN achieves state-of-the-art results, showing its effectiveness across linear and non-linear models and its ability to adapt data distributions for better predictability. Additionally, PTN supports interpretability and transferability, offering potential applications in other time series tasks, like anomaly detection and classification.

**Strengths:**

1. PTN offers a general, model-agnostic approach to improve time series forecasting across diverse datasets.

2. It achieves state-of-the-art results, enhancing accuracy and robustness for both linear and non-linear models.

3. This paper conducts too many experiments to show the effectiveness of their framework.

**Weaknesses:**

1. The paper’s abstract and introduction do not clearly convey the overall research idea and process, making it difficult to understand the framework.

2. The authors mention that PTN shows potential to make time series forecasting more interpretable. However, the enhanced embedding is latent, produced by deep learning. It is unclear how this actually enhances interpretability.

3. The authors provide a caption for the framework, but it does not help clarify the framework’s procedure. It's unclear how the transformed embedding is involved in enhancing the prediction task. I suggest that the authors include a framework overview to illustrate the entire process.

**Questions:**

See Weaknesses

---

> ### Author Response · Authors · 2024-11-21
> **Response to W1, W2 and W3**
>
> > [W1-The paper's abstract and introduction do not clearly convey the overall research idea and process, making it difficult to understand the framework.]
>
> To put it briefly, the core idea of our paper is to **learn an effective transformation of raw data** to enhance a forecasting model's performance compared to using the original data. Unlike traditional methods that rely on handcrafted data transformation techniques such as instance normalization, data augmentation, and preprocessing like down-sampling and patching, we propose the use of a learnable neural network. This network, trained in an end-to-end manner, is termed the Proximal Transformation Network (PTN). As the name suggests, PTN aims to generate transformed data closely resembling the original while enabling improved model performance. The proximity loss acts as a regularization term to reinforce the generalization on the raw data. Additionally, predictability is assessed using a standard loss, like Mean Squared Error (MSE), computed between the label and the model's predictions, both derived from the PTN-generated transformed data. The motivation of PTN is intuitive, but how to learn such an effective transformation is non-trivial.
>
> In our revised paper, we have now made clearer explanations on these aspects in the 2nd and 3rd paragraphs of the Introduction. Moreover, we've revised Figure 3 with workflow markers to aid in understanding the framework. Please also see our detailed response to W3 below.
>
> > [W2-The authors mention that PTN shows potential to make time series forecasting more interpretable. However, the enhanced embedding is latent, produced by deep learning. It is unclear how this actually enhances interpretability.]
>
> We had discussions on the interpretability of our PTN model in Section 5.2 "Interpretation of Proximal Transformation", focusing on the following two aspects.
>
> 1. In Figure 4, we demonstrated that the transformed data generated by PTN show distinct clustering patterns in the loss space, which can be correlated to the predictability of the tested time series. We would like to clarify **such visual analysis is performed on the raw and transformed data**, without involving the latent embedding space.
> 2. In Figure 5, we depicted how all transformed time series evolve to resemble the raw time series during training (see 'raw' and 'transformed' in Figure 5 (a) and (c)). Again, the interpretability here is discussed based on raw and transformed data, not embeddings. The only instance involving "the latent embedding" is in Figure 5 (d), where we examine the efficacy of the employed Convolution Encoder. By using the same "Point-wise Linear Head" to decode the Encoder's output for each variate, we confirmed that the Convolution Encoder manages to align the distribution of different variates, in comparison with RevIN shown in Figure 5 (b). **This analysis examines the utility of the intermediate component of our model, specifically the Encoder.**
>
> Overall, our interpretability analysis primarily focuses on the data before and after transformation. We will make this point clearer in the revision.
>
> > [W3-The authors provide a caption for the framework, but it does not help clarify the framework's procedure. It's unclear how the transformed embedding is involved in enhancing the prediction task. I suggest that the authors include a framework overview to illustrate the entire process.]
>
> Following the suggestion, we have revised the framework in Figure 3. To be specific, we've rearranged the module positions and incorporated arrows with different colors and numbered labels to clearly denote the two sequential steps: proximal transformation and prediction. The caption now includes a detailed description of the entire pipeline to improve clarity. Additionally, we have included a new Figure 7 in Appendix 4.2, which provides a flow chart illustrating the operation of PTN and the predictor during both training and inference processes.
>
> Essentially, the PTN framework functions as an end-to-end data transformation system, generating **transformed time series** for any backbone predictor model to train on. Importantly, **no embeddings are used directly for prediction** with the backbone model.

---

> > ### Comment · Reviewer_t7bj · 2024-11-22
> > **Response to Rebuttal**
> >
> > I have reviewed the authors' rebuttal, and while they have addressed most of my concerns, I will maintain my current score due to the paper's quality, novelty, and presentation.

---

### Official Review · Reviewer_pXSV · 2024-11-03

**Soundness:** 3
**Presentation:** 3
**Contribution:** 3
**Rating:** 6
**Confidence:** 3

**Summary:**

The paper introduces the Proximal Transformation Network (PTN) as a data-centric plugin for enhancing time series forecasting. The proposed PTN aims to find optimal data transformations that improve model performance while preserving data fidelity. Extensive experiments demonstrate state-of-the-art results when the method is integrated with various forecasting models. The key contributions include a reformulation of the time series forecasting problem, the introduction of PTN, and successful performance on seven real-world datasets. The approach highlights the potential of data-centric methods in advancing time series forecasting research

**Strengths:**

1.The paper is well-organized and easy to understand.
2.The proposed model can be widely applied.
3.The proposed model achieves SOTA performance.

**Weaknesses:**

1.In section 3.2, while two losses are considered, what are the motivations/insights of the losses. The reason why they have an influence on the results should be explained.
2.As a plug-and-play model, whether it is lightweight and easy to use is an important criterion, but the experiment does not analyze the time and space complexity of the proposed model.
3.While the performance of the model is not promising enough, the authors don’t analyze the results or explain the pattern.

**Questions:**

1.What is the motivation of losses in section 3.2?
2.Please study the time and space complexity of the proposed model.
3.Why the performance is not stable compared with baselines?

---

> ### Author Response · Authors · 2024-11-21
> **Response to W1 and W2**
>
> > [W1-What is the motivation of losses in section 3.2? ]
>
> In short, we question **if the raw data are "good" enough to train a TSF model**. If not, **how can we find more data with higher chances of being "good" for a model?** Specifically, we use **$l_{pred}$ to measure how good the data are** and **$l_{prox}$ to guide the search for improved data.**
>
> Noise in time series data is common and often unidentifiable. When a model encounters high loss on certain samples, it is unclear whether the issue lies in the model’s inability to learn patterns or the presence of noise. **If the noise could be removed, the model would learn better.** Therefore, we want to shift the original time series to see if the model can fit the data better and hopefully predict better. As shown in Figure 2, we begin with arbitrary data transformations and gradually approach the original data, effectively moving along $l_{prox}$ from larger to smaller values. During the process of "moving along the axis", we have different sets of data (in theory we have infinite sets). We can train a predictor to see if it can fit the data well on the current position on this axis by measuring their performance with $l_{pred}$, which indicates how they predict on the transformed data. This way, we gradually approach the transformed series and simultaneously train our predictor. Figure 2 (b) validates our assumption that a predictor can learn better with transformed data. In vanilla training, with fixed $l_{prox}$ as 0, the predictor can only learn suboptimal results. Whereas our method relaxes the constraints on the objective (from a sample $y$ only to a set of $\tilde{y}$ proximal to $y$) to train better and more robust predictors.
>
> We will emphasize the motivation and make relevant presentations clearer.
>
> > [W2-Please study the time and space complexity of the proposed model. ]
>
> We have added a "Complexity Analysis" section in **Appendix A.4.3** to discuss the complexity of PTN. By analyzing the core components of PTN, intra-patch and channel-wise attentions, we identified batch size as a previously overlooked factor in complexity analysis (see P1 in Appendix A.4.3). This led us to present more efficient implementations (see P2 and P3 in Appendix A.4.3) that limit parallel computations within a batch. Experimental results (Table 8, Table 9, Figure 10, Figure 11 in the revised paper) demonstrate these implementations reduce PTN's complexity to **match that of linear models**, enhancing its practicality.
>
> These efficient implementations and the corresponding empirical studies are summarized as follows:
>
> - **Mask Fusion** (P2 in Appendix A.4.3): We introduced a method to accelerate attention computations by merging intra-patch and channel-wise attention masks. This reduced complexity from $\mathcal{O}((l_p + C)LC)$ to $\mathcal{O}(l_p^2C^2)$, where $l_p$ (patch length) is as small as 4 in our experiments. Details and results supporting this method are in Appendix A.4.3, Figure 8, and Figure 9, showing its effectiveness in improving PTN's efficiency. A brief view of the results is shown in table **T1 in "Comments to All"** and in Figure 9 in Appendix 4.3.
> - **Variate Sampling** (P3 in Appendix A.4.3): According to our new implementation, adopting the variate sampling technique proposed in iTransformer [1] can also lead to improvement in terms of both training and inference efficiency. The relative increase in time and memory costs can be reduced from several times to a factor of **less than one**. The results are reported in table **T2 in "Comments to All"** and in Tables 8 and 9 in Appendix A.4.3.
>
> In addition to performance-neutral optimizations introduced above, there are also **lossy acceleration methods**. As discussed in Section 5.3, a data transfer method can improve training efficiency by avoiding the direct training of a complex backbone. For inference, we propose a method based on train-time distillation that generates a student model without PTN, incurring no additional inference overhead but requiring only the training of an extra backbone. Details are provided in Appendix A.4.4, along with experimental results shown below. However, this approach is not universally applicable and may fail in certain cases, such as the iTransformer on ETT datasets, for reasons outlined in response to W2. The brief results are shown in **T3 in "Comments to All"** and full results are shown in Table 10 in Appendix 4.4.
>
> [1] iTransformer: Inverted Transformers Are Effective for Time Series Forecasting

---

> ### Author Response · Authors · 2024-11-21
> **Response to W3**
>
> > [W3-Why the performance is not stable compared with baselines?]
>
> The results are now shown in Table 13. We have also observed the differences in performance boosts from diverse angles. We are going to organize our analysis from aspects of backbones and datasets.
>
> **Varying performances on different backbones**
>
> As the reviewer pointed out, the proposed PTN module is more effective for linear models while having varying boosts for complex models. This is because of some of the intrinsic drawbacks in transformer-based models. As the paper [2] suggests, the transformers (both temporal and channel-wise) suffer from overfitting problems, especially on small datasets. In Section 4.1, the paragraph starting with **"Training loss"** explains that we do not use a strict constraint on gradient norms for consideration of convergence. Thus the revised loss does not guarantee reaching Pareto frontier and results in unstable performance for more complex models. For better generalization on more models and more stable performance, we plan to explore it in future work.
>
> [2] SAMformer: Unlocking the Potential of Transformers in Time Series Forecasting with Sharpness-Aware Minimization and Channel-Wise Attention
>
> **Varying performances on different datasets**
>
> Another observation is that the proposed models do not constantly improve performances over different datasets, we believe some of the cases are occasional and we add additional experiments on PeMS datasets as reviewer gMsX suggested. The detailed experiments on PeMS datasets are shown in Table 14 in the Appendix A.6, which shows the effectiveness of our methods on complex datasets. We also want to point out that the results in the iTransformer [1] paper are mistakenly described. The Github issue (https://github.com/thuml/iTransformer/issues/91) serves as a proof as well as other discussions that can not reproduce the PeMS results (https://github.com/thuml/iTransformer/issues?q=is%3Aissue+pems). We have rerun the baseline for more fair comparisons.
>
> For the current work, we believe the instability is acceptable **because this does not interfere with the improvement of SOTA methods on each dataset**. For ETT datasets where linear models are better, the improvement is stable, and for the rest three datasets where transformer-based methods excel, we also have fair improvements.
>
> We will add these explanations to our paper.

---

### Official Review · Reviewer_CnMd · 2024-11-04

**Soundness:** 2
**Presentation:** 2
**Contribution:** 2
**Rating:** 5
**Confidence:** 4

**Summary:**

This paper proposes a Proximal Transformation Network to learn effective transformations while maintaining proximity to the raw data to ensure fidelity. The model includes a convolution-based Encoder and an attention-based Encoder that provide transformation on different levels of proximity. The training involves a co-optimization of the proximity of the transformed data and forecasting accuracy. The method achieves state-of-the-art performance on seven real-world datasets. Additionally, the paper shows that the proximal transformation process can be interpreted in terms of predictability and distribution alignment among channels.

**Strengths:**

* The traditional time series prediction task has been redefined as a "two-step problem," the goal is to learn predictions on the transformed data and align them with the raw series, showcasing innovation.

* The proposed method achieves state-of-the-art performance on seven real-world datasets. The ablation experiments are comprehensive.

* The effectiveness of the proposed module is demonstrated through the distribution of data on the loss surface, revealing its ability to categorize time series into predictable and unpredictable groups in a self-supervised manner, with a particular focus on enhancing performance for the former.

**Weaknesses:**

* When the predictor is a linear model, the complexity of PTN seems to far exceed that of the predictor itself. It is suggested to provide the time and memory complexity analysis of the proposed module. Additionally, for predictors that include modules capturing channel-wise and patch-wise correlations, the proposed PTN appears redundant, which may affect its generalizability.

* Based on the results in Table 10, the PTN module appears to enhance performance primarily for simple linear models, while its effectiveness on more complex models, such as iTransformer and PatchTST, varies against the dataset. Considering that the main objective of this paper is to propose a general plugin, it is essential to select a sufficient range of predictors for experimentation.

* The article contains several errors that require careful proofreading. For example, "Encoder" in line 65 should be corrected to "Decoder," and the shape of the matrix in line 129 needs clarification. Additionally, there are concerns regarding Figure 2(b), where $l_{\text{raw}}$ decreases as $l_{\text{pred}}$ increases, which seems counterintuitive and requires further explanation.

**Questions:**

See Weaknesses.

---

> ### Author Response · Authors · 2024-11-21
> **Reponse to W1**
>
> > [W1-When the predictor is a linear model, the complexity of PTN seems to far exceed that of the predictor itself. It is suggested to provide the time and memory complexity analysis of the proposed module. Additionally, for predictors that include modules capturing channel-wise and patch-wise correlations, the proposed PTN appears redundant, which may affect its generalizability.]
>
> Below, we address the concerns regarding complexity and redundancy through comprehensive theoretical analysis and empirical validation.
>
> **Complexity**
>
> We have added a "Complexity Analysis" section in **Appendix A.4.3** to discuss the complexity of PTN. By analyzing the core components of PTN, intra-patch and channel-wise attentions, we identified batch size as a previously overlooked factor in complexity analysis (see P1 in Appendix A.4.3). This led us to present more efficient implementations (see P2 and P3 in Appendix A.4.3) that limit parallel computations within a batch. Experimental results (Table 8, Table 9, Figure 10, Figure 11 in the revised paper) demonstrate these implementations reduce PTN's complexity to **match that of linear models**, enhancing its practicality.
>
> These efficient implementations and the corresponding empirical studies are summarized as follows:
>
> - **Mask Fusion** (P2 in Appendix A.4.3): We introduced a method to accelerate attention computations by merging intra-patch and channel-wise attention masks. This reduced complexity from $\mathcal{O}((l_p + C)LC)$ to $\mathcal{O}(l_p^2C^2)$, where $l_p$ (patch length) is as small as 4 in our experiments. Details and results supporting this method are in Appendix A.4.3, Figure 8, and Figure 9, showing its effectiveness in improving PTN's efficiency. A brief view of the results is shown in table **T1 in "Comments to All"** and in Figure 9 in Appendix 4.3.
> - **Variate Sampling** (P3 in Appendix A.4.3): According to our new implementation, adopting the variate sampling technique proposed in iTransformer [1] can also lead to improvement in terms of both training and inference efficiency. The relative increase in time and memory costs can be reduced from several times to a factor of **less than one**. The results are reported in table **T2 in "Comments to All"** and in Tables 8 and 9 in Appendix A.4.3.
>
> In addition to performance-neutral optimizations introduced above, there are also **lossy acceleration methods**. As discussed in Section 5.3, a data transfer method can improve training efficiency by avoiding the direct training of a complex backbone. For inference, we propose a method based on train-time distillation that generates a student model without PTN, incurring no additional inference overhead but requiring only the training of an extra backbone. Details are provided in Appendix A.4.4, along with experimental results shown below. However, this approach is not universally applicable and may fail in certain cases, such as the iTransformer on ETT datasets, for reasons outlined in response to W2. The brief results are shown in **T3 in "Comments to All"** and full results are shown in Table 10 in Appendix 4.4.
>
> [1] iTransformer: Inverted Transformers Are Effective for Time Series Forecasting
> **Redundancy**
>
> In brief, we argue that the designs in PTN and the predictor are not redundant, as they optimize parameters using different losses: $l_{prox} + l_{pred}$ for PTN and $l_{pred}$ for the predictor, making their search space distinct. To address redundancy concerns, we have explored two questions in Section 5.4, "Ablation Study and the PTN Design":
>
> **Q1**: Are the same "redundant" modules effective when used as a predictor?
>
> **A1:** No, they are not. When we adapt these modules into a predictor without transforming input $X$ (merging transformations on $Y$ and predictions), the performance significantly drops, as shown in Table 5 ("ConvPred").
>
> **Q2**: How does performance change when removing the "redundant" designs?
>
> **A2:** As shown in Figure 6 in the revised paper (note: mislabeled figures have been corrected), the impact of attention is minimal, and the choice of attention depends more on datasets than backbones.
>
> While there may be side effects, such as reduced generalizability due to similar architectures between PTN and backbones, **we cannot conclude the redundancy based on the evidence provided**. We will further explore this issue in the future.

---

> ### Author Response · Authors · 2024-11-21
> **Response to W2 and W3**
>
> > [W2-Based on the results in Table 10, the PTN module appears to enhance performance primarily for simple linear models, while its effectiveness on more complex models, such as iTransformer and PatchTST, varies against the dataset. Considering that the main objective of this paper is to propose a general plugin, it is essential to select a sufficient range of predictors for experimentation.]
>
> The results are now shown in Table 13 in the revised paper. We have also observed the differences in performance boosts from diverse angles. We are going to organize our analysis from aspects of backbones and datasets.
>
> **Varying performances on different backbones**
>
> As the reviewer pointed out, the proposed PTN module is more effective for linear models while having varying boosts for complex models. This is because of some of the intrinsic drawbacks in transformer-based models. As the paper [2] suggests, the transformers (both temporal and channel-wise) suffer from overfitting problems, especially on small datasets. In Section 4.1, the paragraph starting with **"Training loss"** explains that we do not use a strict constraint on gradient norms for consideration of convergence. Thus the revised loss does not guarantee reaching Pareto frontier and results in unstable performance for more complex models. For better generalization on more models and more stable performance, we plan to explore it in future work.
>
> [2] SAMformer: Unlocking the Potential of Transformers in Time Series Forecasting with Sharpness-Aware Minimization and Channel-Wise Attention
>
> **Varying performances on different datasets**
>
> Another observation is that the proposed models do not constantly improve performances over different datasets, we believe some of the cases are occasional and we add additional experiments on PeMS datasets as reviewer gMsX suggested. The detailed experiments on PeMS datasets are shown in Table 14 in the Appendix A.6, which shows the effectiveness of our methods on complex datasets. We also want to point out that the results in the iTransformer [1] paper are mistakenly described. The Github issue (https://github.com/thuml/iTransformer/issues/91) serves as a proof as well as other discussions that can not reproduce the PeMS results (https://github.com/thuml/iTransformer/issues?q=is%3Aissue+pems). We have rerun the baseline for more fair comparisons.
>
> For the current work, we believe the instability is acceptable **because this does not interfere with the improvement of SOTA methods on each dataset**. For ETT datasets where linear models are better, the improvement is stable, and for the rest three datasets where transformer-based methods excel, we also have fair improvements.
>
> We will add these explanations to our paper.
> > [W3-The article contains several errors that require careful proofreading. For example, "Encoder" in line 65 should be corrected to "Decoder," and the shape of the matrix in line 129 needs clarification. Additionally, there are concerns regarding Figure 2(b), where lraw decreases as lpred increases, which seems counterintuitive and requires further explanation.]
>
> Thank you for pointing out these issues. We have carefully proofread the article and corrected the mentioned errors.
>
> Regarding the concerns about Figure 2(b) and the relationship between $l_{raw}$ and $l_{pred}$, we appreciate the opportunity to clarify. Figure 2(b) illustrates that $l_{raw}$ represents prediction results measured on raw data, which are important but not directly used in training due to the transformation. On the other hand, $l_{pred}$ reflects prediction errors on the transformed data, and these two metrics do not necessarily exhibit a positive correlation. In fact, $l_{raw}$ is positively correlated with the sum of $l_{pred}$ and $l_{prox}$. As shown in Figure 2(b), when this sum cannot be further optimized, there exists an optimal allocation of the two losses that minimizes $l_{raw}$. We will revise the explanation in Figure 2 to ensure clarity and avoid any potential misunderstanding. Thank you again for the feedback.

---

> > ### Comment · Reviewer_CnMd · 2024-11-23
> > **Response to author's rebuttal**
> >
> > Thanks for offering the additional complexity analysis and further clarification. Given its current presentation, technical contributions, and effectiveness, I will keep my score.

---

### Author Response · Authors · 2024-11-21
**Comments to All (Part 1)**

We appreciate the constructive comments from reviewers, which have led to significant updates in our paper. The revised parts in the **newly uploaded version** have been highlighted with blue colors for clarity.

For the convenience of the reviewers, we post here the experimental results that more than one reviewer concerns about. Reviewers can find corresponding results associated with our responses according to the labels (e.g. **T1**, **T2**).

**T1** Efficiency improvement with attention mask fusion,  hyper-parameters are set as batch size = 128, and number of variates = 64. The time complexity is measured by execution time in ms/iteration and memory complexity is measured by memory consumption in GB.

|           |                  | Execution Time(**ms/iter**) | Memory Consumption(**GB**) |
| --------- | ---------------- | --------------------------- | -------------------------- |
| **train** | w. mask fusion   | 163.6661                    | 9.40918                    |
|           | w/o. mask fusion | 213.2196                    | 5.612305                   |
| **infer** | w. mask fusion   | 337.8378                    | 2.793945                   |
|           | w/o. mask fusion | 467.2897                    | 2.331055                   |

**T2** Training efficiency improvement with variate sampling. The number of variates sampled is 64. The time complexity is measured by execution time in ms/iteration and memory complexity is measured by memory consumption in GB. The additional cost of our PTN module is measured by the relative increase compared to backbone models in "inc(%)". Since the fused attention can be also applied to inter-patch attention, we observe speed-up for PatchTST.
|         |                |    PTN-DLI    |           |             |          |    PTN-iTr    |           |            |          |    PTN-Pat    |           |            |           |
| :-----: | :------------: | :-----------: | :-------: | :---------: | :------: | :-----------: | :-------: | :--------: | :------: | :-----------: | :-------: | :--------: | :-------: |
|         |                | time(ms/iter) |  inc(%)   | memory (GB) |  inc(%)  | time(ms/iter) |  inc(%)   | memory(GB) |  inc(%)  | time(ms/iter) |  inc(%)   | memory(GB) |  inc(%)   |
| traffic | w./o. sampling |    150.105    |  710.45   |    6.830    |  563.58  |    149.201    |   82.51   |   7.928    |  62.57   |    79.701     |   31.56   |   3.330    |   33.38   |
|         |  w. sampling   |    10.775     | **51.92** |    0.042    | **3.83** |     9.383     | **36.56** |   0.033    | **2.48** |     4.779     | **15.74** |   -0.139   | **-8.64** |
| weather | w./o. sampling |     9.452     |   86.11   |    0.070    |   8.66   |     4.301     |   30.15   |   0.039    |   3.48   |     4.429     |   24.87   |   -0.137   |   -9.84   |
|         |  w. sampling   |     2.449     | **28.30** |    0.035    | **8.39** |     1.941     | **18.40** |   0.023    | **3.90** |    -0.690     | **-5.28** |   -0.008   | **-1.02** |

**T3** Performance of cost-free student model inference. We use a look-back window with the length of 96 and horizon lengths in {96, 192, 336, 720}. The results are averaged on different horizons. The full results are shown in Table 10 in Appendix A.4.4.

|                  | Dlinear |       | +PTN(stu) |           | PatchTST |           | +PTN(stu) |           | iTrasnformer |           | +PTN(stu) |           |
| ---------------- | ------- | ----- | --------- | --------- | -------- | --------- | --------- | --------- | ------------ | --------- | --------- | --------- |
| metrics          | MSE     | MAE   | MSE       | MAE       | MSE      | MAE       | MSE       | MAE       | MSE          | MAE       | MSE       | MAE       |
| ETTh1(avg)       | 0.446   | 0.434 | **0.444** | **0.430** | 0.469    | **0.455** | **0.460** | **0.455** | **0.454**    | **0.448** | 0.498     | 0.479     |
| ETTh2(avg)       | 0.374   | 0.399 | **0.371** | **0.396** | 0.387    | 0.407     | **0.386** | 0.410     | **0.383**    | **0.407** | 0.437     | 0.440     |
| electricity(avg) | 0.219   | 0.298 | 0.222     | **0.296** | 0.205    | 0.290     | **0.199** | **0.281** | 0.178        | 0.270     | **0.169** | **0.261** |
| traffic(avg)     | 0.627   | 0.378 | 0.639     | **0.369** | 0.481    | 0.300     | 0.491     | **0.286** | 0.428        | 0.282     | 0.442     | **0.274** |

---

### Author Response · Authors · 2024-11-21
**Comments to All (Part 2)**

**T4** Performance of PeMS datasets. We use a look-back window with the length of 96 and horizon lengths in {12, 24, 36, 48}. The results are averaged on different horizons. The full results are shown in Table 14 in Appendix A.6. We achieve comparable performance improvement except for iTransformer, as it is more difficult to converge and easier to overfit.

|             | Dlinear |       | \+PTN     |           | PatchTST |       | \+PTN     |           | iTransformer |           | \+PTN     |           |
| ----------- | ------- | ----- | --------- | --------- | -------- | ----- | --------- | --------- | ------------ | --------- | --------- | --------- |
| metrics     | MSE     | MAE   | MSE       | MAE       | MSE      | MAE   | MSE       | MAE       | MSE          | MAE       | MSE       | MAE       |
| PEMS03(avg) | 0.317   | 0.364 | **0.172** | **0.277** | 0.298    | 0.368 | **0.123** | **0.230** | 0.113        | 0.222     | **0.102** | **0.207** |
| PEMS04(avg) | 0.329   | 0.377 | **0.193** | **0.292** | 0.341    | 0.411 | **0.153** | **0.257** | **0.111**    | **0.221** | 0.121     | 0.225     |
| PEMS07(avg) | 0.319   | 0.368 | **0.182** | **0.283** | 0.293    | 0.366 | **0.120** | **0.218** | **0.101**    | 0.204     | **0.101** | **0.194** |
| PEMS08(avg) | 0.321   | 0.372 | **0.259** | **0.308** | 0.299    | 0.372 | **0.200** | **0.252** | **0.150**    | 0.226     | 0.173     | **0.220** |

---

### Author Response · Authors · 2024-11-27

Dear reviewers：

Thank you all for your active participation in the discussion phase and for providing valuable and constructive feedback. We have undertaken extensive revisions in response to the reviews. For a brief summary, we would like to list our key revisions below to facilitate further discussions.

**Summary of the Revisions**

1. (Suggested by Reviewers CnMd, pXSV, gMsX) We have expanded Section A.4.3 with a comprehensive 3-page analysis on the computational complexity of PTN, as well as general attention mechanisms within time series transformers. By introducing an attention fusion mask and variable sampling, we significantly enhance both training and inference efficiency. Incorporating these methods, the complexity of PTN has been reduced to a level no higher than that of the backbone model.
2. (Inspired by Reviewer gMsX) To address concerns regarding complexity, in Section A.4.4, we present an alternative implementation option: a train-time distillation version of PTN. This variant allows for inference at zero additional cost, albeit with a minor compromise in performance.
3. (Suggested by Reviewer gMsX) Extensive experiments on PeMS datasets, newly included in Table 14, demonstrate the effectiveness of our approach on complex datasets.
4. (Suggested by Reviewer t7bj) An additional section, A.4.2, has been added to better illustrate the overall pipeline of our method. Additionally, Figure 3 has been revised for a clearer presentation of the architecture.
5. The paper has undergone thorough proofreading, correcting previous typos and clarifying any ambiguous points.
We believe these revisions can enhance our paper in terms of soundness, presentation as well as contribution.

**Looking Forward to Further Discussions**

We hope that further discussions will be made, particularly regarding the core emphasis of our work. At the heart of our paper is to question the effectiveness of current time series datasets. Our approach involves transforming the original data to investigate whether models can learn more effectively from this transformed data. The conclusion is that, by not relying solely on raw data, models often achieve more accurate predictions even when applied back to the raw data. As is widely acknowledged, the scaling law encompasses three dimensions: computation, parameter size, and data size. While much research focuses on scaling parameters, our work offers a new perspective on scaling data, specifically in terms of quality [1,2]. Although the end-to-end data transformation process is straightforward and admittedly still evolving, we successfully feed models with improved data without resorting to heuristics, even under simple designs. We sincerely hope that reviewers will engage in deeper discussions with us during the prolonged rebuttal phase and provide valuable feedback. We would greatly appreciate it if reviewers could reconsider our work from a fresh angle.

[1] ScalingFilter: Assessing Data Quality through Inverse Utilization of Scaling Laws

[2] Scaling Parameter-Constrained Language Models with Quality Data

Best regards!

---

### Meta-Review · Area_Chair_Q82q · 2024-12-22

**Metareview:**

This paper introduces the Proximal Transformation Network (PTN), a data-centric plugin for enhancing time series forecasting. PTN is designed to optimize data transformations while maintaining proximity to raw data, with the goal of improving forecasting accuracy and interpretability. The proposed framework integrates a convolutional encoder and an attention-based decoder to generate transformed data for model training.

The reviewers agreed that this paper offers a novel perspective by redefining the time series forecasting problem as learning data transformation and predicting on transformed data. Reviewers had concerns about the computational complexity of the proposed PTN methods, while the authors managed to address this issue well during the rebuttal. Although the paper presents a promising concept, reviewers still have concerns about motivation, technical quality, presentation, and experimental results after the rebuttal. As such, I am inclined to recommend rejecting this paper.

**Additional Comments On Reviewer Discussion:**

During the rebuttal, 3 out 4 reviewers responded to the authors’ replies. Reviewer CnMd kept the original score due to concerns about presentation, technical contributions, and effectiveness. Reviewer t7bj also kept the original score due to the paper's quality, novelty, and presentation. Reviewer gMsX increased the score, however, the updated score is still negative due to experimental results and presentation.

Overall, the quality of the paper has improved following the rebuttal. Notably, the authors successfully addressed concerns related to computational complexity and included additional results on larger datasets. However, the reviewers continue to highlight significant issues regarding motivation, technical rigor, presentation clarity, and the consistency of experimental results. As such, the paper still falls below the acceptance threshold for ICLR.

---

### Decision · Program_Chairs · 2025-01-22

Reject